# Immunodiagnostic plasma amino acid residue biomarkers detect cancer early and predict treatment response

Cong Tang [1,2,3] ✉, Patrícia Corredeira[1], Sandra Casimiro [1], Qi Shi[4], Qiwei Han [4], Wesley Sukdao[2], Ana Cavaco[1], Cecília Melo-Alvim[5], Carolina Ochôa Matos[6,7], Catarina Abreu[5], Steven Walsh[2], Gonçalo Nogueira-Costa[5], Leonor Ribeiro[5], Rita Sousa[5], Ana Lorena Barradas [1], João Eurico Fonseca[6,7], Luís Costa [1,5] ✉, Emma V. Yates [2] ✉ & Gonçalo J. L. Bernardes [1,8,9] ✉

The immune response to tumour development is frequently targeted with therapeutics but remains largely unexplored in diagnostics, despite being stronger for early-stage tumours. We present an immunodiagnostic platform to detect this. We identify a panel of amino acid residue biomarkers providing a signature of cancer-specific immune activation associated with tumour development and distinct from autoimmune and infectious diseases, measurable optically in neat blood plasma, and validate within N = 170 participants. By measuring the total concentrations of cysteine, free cysteine, lysine, tryptophan, and tyrosine protein-incorporated biomarkers and analyzing the results with supervised machine learning, we identify 78% of cancers with 0% false positive rate (N = 97) with an AUROC of 0.95. The cancer, healthy, and autoimmune/infectious biomarker pattern are statistically significantly different (p < 0.0001). Smaller-scale changes in biomarker concentrations reveal inter-patient differences in immune activation that predict treatment response. Specific concentration ranges of these biomarkers predict response to Cyclin-dependent kinase inhibitors in advanced breast cancer patients (p < 0.05), identifying 98% of responders (N = 33). Here we provide an immunodiagnostic technology platform that, to our knowledge, has not been previously reported, and prove initial clinical application in a cohort of N = 170, including proof of concept in Multi Cancer Early Detection and personalized medicine.

The idea that the immune system is involved in the surveillance for neoplastic disease originated in the nineteenth century[1]. While enhancing the immune response to tumour development has become standard-of-care in therapeutics[2], detecting the immune response for diagnostic purposes has remained largely unexplored. However, the central and general role of tumour immunosurveillance in cancer onset and progression suggests that detecting tumour immunosurveillance for diagnostics could have significant potential.

Instead, liquid biopsies, which aim to detect cancer presence or prognosis via a non-invasive blood test (Fig. 1a), have become

**Fig. 1 | Detecting tumour immunosurveillance for diagnostics. a** A patient with a breast tumour provides a 5 mL whole blood sample. **b** The c. 70 mg/mL protein component plays roles in immunosurveillance, in which the immune system identifies and eliminates malignant cells. This process increases in early-stage cancer before cancer immunoediting allows immune escape and metastasis. It is also the target of immunosurveillance increasing drugs, such as immune check-point inhibitors (ICI's) and a recently discovered additional mechanism of Cyclin-dependent kinase inhibitors (CDKi's)[36]. **c** Example of immune surveillance enhancement observed upon development of prostate cancer[12]. **d** The host immune response includes up to 3000 additional proteins[16], however, clinically meaningful quantification remains challenging due to high intra and inter-individual variation. **e** We introduce the concept of taking an average across all individual protein targets. This Amino Acid Concentration Signature (AACS) provides a protein cross-section, which reduces intra and inter-individual variation via the Law of Large Numbers.

synonymous with circulating tumour DNA (ctDNA) based assays. Though laudable progress has been made, exemplified in the NHS-Galleri® trial-GRAIL cancer detection studies[3], the primary issue is insufficient signal. The amount of ctDNA shed is a proxy for tumour burden, with early-stage tumours shedding as little as 0-1 copies/mL[4], 76–92% of Stage I cancers are missed[3,5,6]. Predicting treatment response in baseline naïve patients remains challenging for the same reason, and response can only be *observed* when ctDNA starts to rise upon further cancer progression[7,8], introducing a several-month delay into the patient care path. In addition, ctDNA levels are usually prognostic of survival in general, rather than predictive of response to individual treatments[9], and are unlikely to provide a general platform for adjuvant treatment guidance in early-stage cancer patients due to insufficient signals.

In contrast, the immune response to tumour development is stronger for early-stage tumours, with cancer immunoediting ultimately allowing immune escape and metastasis[10,11] (Fig. 1b). Cancer-specific immune surveillance changes include class switching between the immunoglobulin (Ig) subtypes that have been observed to substantially change their total concentrations in patient blood plasma, as shown for the development of prostate cancer[12] (Fig. 1c). Several fold changes in the total level of IgA, IgG, IgE, IgD, and IgM antibody proportions have been observed in cancers such as breast, colorectal, pancreatic, and prostate. These are different to the immunological

pattern changes observed in non-cancerous immune activation due to autoimmune diseases[13] and infection[14,15].

The host immune response to tumour development includes not only immunoglobulins themselves but additionally alternations to approximately 1700 of 3000 host proteins present within plasma[16] (Fig. 1d). Proteomic measurement of these alterations has remained challenging due to high intra and inter-individual variability due to factors such as affinity/avidity convolution, substrate binding effects, low antibody half-life in plasma, and proteomic batch effects. Intrinsically patient total protein concentration is known to vary substantially between 60–80 mg/mL, impacting results[17].

In this work, we use a machine-learning inspired approach to redefine the immunosurveillance targets away from the individual protein targets themselves and towards their common building blocks (Fig. 1e), which we prove that we reliably quantify within neat patient blood plasma in $N = 170$ patient plasma samples. We then establish clinically multi-cancer early detection, and prediction of baseline naïve

patient response to Cyclin-dependent kinase inhibitors (CDKi's), which are revolutionising the treatment of breast cancer patients.

## Results

### Conceptual design of immunodiagnostic biomarkers

Blood plasma contains up to 5000 proteins[18] and can therefore be thought of as a high-dimensional dataset. When measured individually, these proteins vary substantially among patients of the same disease state[18]. High dimensionality is known to make accurate predictions difficult, because the number of individuals must exceed the number of predictors by several fold. In domains such as natural language processing and image processing, where high dimensionality is common, embeddings are employed to reformulate the dataset. An embedding captures the essence of data by representing it as points within an n-dimensional space. This vector representation ensures that data points with similarities are grouped closely together (Fig. 2a)[19]. Popular examples of this approach include word2vec and Google's search by

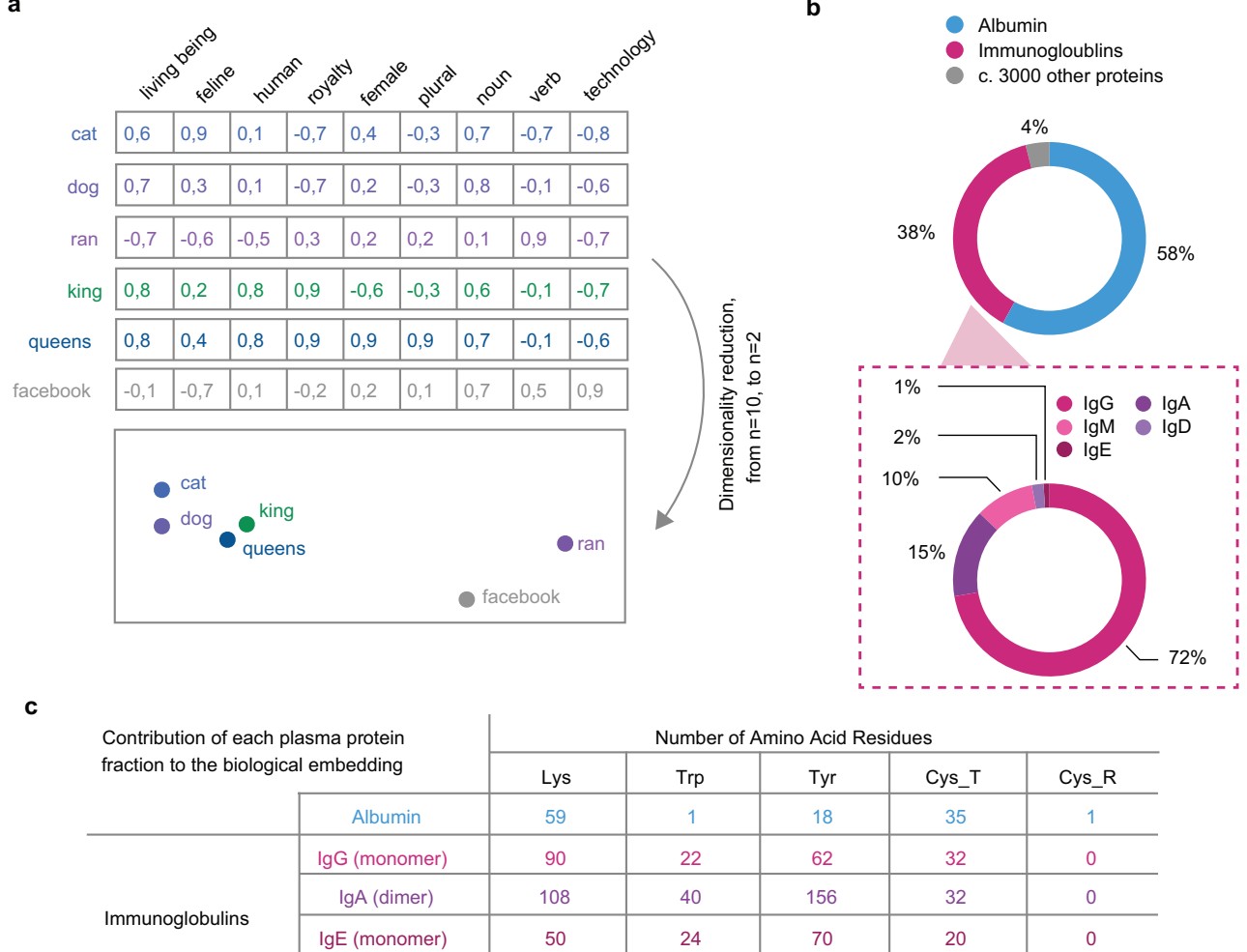

Fig. 2 | Design of biomarkers using a biological embedding. a Embeddings are frequently used to process complex systems with many targets for machine learning applications, such as in natural language processing. Such embeddings represent each target as a vector of component parts within a high-dimensional space, and combines the component vectors in a dimensionality reduction step, such that similar targets cluster together in reduced n-dimensional space.
b Contrary to traditional embeddings that derive the relative contributions of the dataset dimensions to maximise captured variation, our biological embedding is determined by the proportional molar concentration of components in blood plasma. The Immunoglobulin fraction, c. 38% of proteins, is further divided into immunoglobulin classes, with the proportional contribution shown in healthy individuals. c We performed bioinformatic analysis to determine the optimal biological embedding dimensions to capture this variability. Possible dimensions include any of the $N = 20$ major amino acid types, which comprise the major plasma protein fractions. We identified $N = 5$ amino acid types whose numbers changed substantially across the proportion-weighted fractions, such that they would be detectable in an average biological embedding signature.

image functionality, where words or images with similar meanings or content are clustered near each other in the embedded space[20].

We hypothesised that the plasma proteome, including the immunodiagnostic immunoglobulin component (Fig. 2b), could be redescribed utilising an intrinsic biological embedding with amino acid residues as component parts. Because amino acid residues, rather than amino acid metabolites free in solution, were the targets of our investigation, we performed bioinformatics analysis considering the canonical protein sequences of the major plasma protein classes, and identified several key amino acids – lysine (Lys), tryptophan (Trp), tyrosine (Tyr), cysteine (Cys_T), and cysteine not engaged in disulphide bonds (Cys_R), whose numbers changed significantly across these classes (Fig. 2c and Supplementary Fig. 1 and 2). If it were possible to quantify the total levels of these amino acid residues in neat patient blood plasma, then the immunosurveillance changes which have been previously observed specific to individual cancer types (such as shown in Fig. 1c) would be conveniently detectable, with a signal strength 100 trillion times stronger[4] than ctDNA based approaches.

## Measuring immunodiagnostic biomarkers in neat patient blood plasma

An optical detection strategy enables high-throughput solution-phase measurements. Labelling our amino acid targets enables optical detection. To label our targets while enabling efficient and reproducible sample processing, we developed a three-fold strategy (Fig. 3a).

The first component of our strategy is using bioorthogonal labels that are targeted to react exclusively with the side-chains (R-groups) of amino acid types to which they are targeted, and not with other amino acid types or components of the plasma solution. Second, we use fluorescence-generating ("fluorogenic") labels. The use of labels that are not fluorescent until after they undergo a covalent reaction with their specific targets can obviate the requirement to remove unreacted labels from neat patient blood plasma. Third, fluorescence from the reacted label should be able to be quantitatively related to the concentration of the targeted species, so that the fluorescence measured from our average signature-based approach is proportional to the changing fractional composition of blood plasma. We have described[21–24] quantitative labelling of amino acid targets within dilute solutions of individual proteins. See Supplementary Fig. 3 for label chemical structures and reaction schema.

However, to determine whether the labels could be directly added to neat patient blood plasma without a purification step, we had to account for the impact of autofluorescence, which can confound fluorescence assays[25]. We performed two-dimensional excitation and emission scans to verify that autofluorescence from unreacted neat blood plasma was minimal at our chosen excitation and emission wavelengths (Fig. 3b). We used the theoretical total protein concentration range of patient blood plasma to identify dilutions which placed the total concentration of the determined amino acid type within the quantitative range of each of our labelling reactions, and verified experimentally (Supplementary Fig. 4). Collectively, these steps ensured that any auto fluorescent background could simply be subtracted from the labelled patient plasma samples.

We were then able to use the relationship between the fluorescence intensity measured from solutions of known amino acid concentration (known protein concentration times known number of targeted amino acid R-groups within the protein sequence) to derive a calibration curve that we could use to transform the raw measured fluorescence intensities of our neat blood plasma solution into their corresponding amino acid concentrations. Calibration curves are shown in Fig. 3c and Supplementary Fig. 4. This provides each patient with an *Amino Acid Concentration Signature* (AACS), which represents an embedding over all plasma proteins in n-dimensional amino acid space.

Finally, we tested whether the AACS measured experimentally matched the AACS calculated theoretically from the known concentrations of individual plasma proteins in healthy patient blood plasma. To do this, we calculated a theoretical AACS using a weighted average of the plasma protein amino acid sequences and their proportional molar concentration within blood plasma using existing proteomic data from the Human Protein Atlas and compared the results to our experimental measurements (Fig. 3d). Agreement between the results suggests that measuring an average amino acid concentration signature in neat patient blood plasma reflects the fractional composition of patient plasma proteins, and further verifies that direct measurement of amino acid concentrations in unseparated, neat patient blood plasma is feasible. This conveniently eliminates the need for laborious separation or enrichment steps in traditional proteomic assays.

## Protein Cross Section Changes Driven by Immune Response

To test the hypothesis that the immune system is the main mediator of changes to AACS, we explored whether we could detect the difference between immunologically hot and immunologically cold tumours. To test this at the proof-of-concept level, we constructed a mouse model of hot and cold tumours[26] and showed that these produce distinct AACS, as shown in Fig. 3e. This confirmed our hypothesis that changes in the host immune response drive AACS biomarker changes.

## Multi cancer early detection (MCED)

Breast, colorectal, pancreatic, and prostate cancers are prevalent cancers in which significant immunoglobulin class switching has been observed compared to cancer-negative controls[14]. Therefore, we chose this patient group as initial validation of our MCED concept. To test our hypothesis that the AACS of patients with cancer would differ from the AACS of patients without cancer, we measured plasma samples collected retrospectively from $N = 77$ patients followed at Hospital de Santa Maria in Lisbon, Portugal. $N = 20$ samples were collected from healthy donors. The samples from patients with cancer were diagnostic samples, collected prior to the patient starting treatment. We did not exclude comorbid conditions or infections from the cancer-positive or negative groups. Additional patient clinical and demographic characteristics are included in Supplementary Data 1. Approximately 5 mL non-fasting blood samples were collected and processed into plasma using standard procedures (see Materials & Methods).

We obtained a 500 μL plasma sample from each patient, diluted each plasma sample 1:60 in PBS buffer, and then added 100 μL to each of four microplate wells (see Materials & Methods). We added one labelling solution to each of the four wells to label the indicated amino acid type. We measured the resulting fluorescence from each well using the excitation and emission wavelengths described in Supplementary Table 1, subtracted the background from the unreacted labelling solution and the diluted patient plasma sample, and used the calibration curves shown in Fig. 3c and Supplementary Fig. 4 to convert the measured fluorescence intensities into amino acid concentrations. We performed the measurements in triplicate and calculated the mean of each triplicate measurement and plotted the results in Fig. 4a.

The AACS distributions of $N = 20$ cancer negative patients and $N = 77$ cancer positive patients were visually distinct, as shown in Fig. 4a. To determine whether the AACS distributions of the cancer negative and cancer positive patients were statistically distinct, we performed a Multidimensional Analysis of Variance (MANOVA) statistical test. A MANOVA tests whether the independent grouping variable (cancer status) simultaneously explains a statistically significant amount of variance in the dependent variables (the amino acid concentration measurements of each patient's AACS signature). We rejected the null hypothesis that the cancer negative and cancer positive AACS distributions were identical at the 1% significance level ($p = 3.11 \times 10^{-14}$), proving that the distributions were statistically distinct. We further analysed the contributions of each individual amino

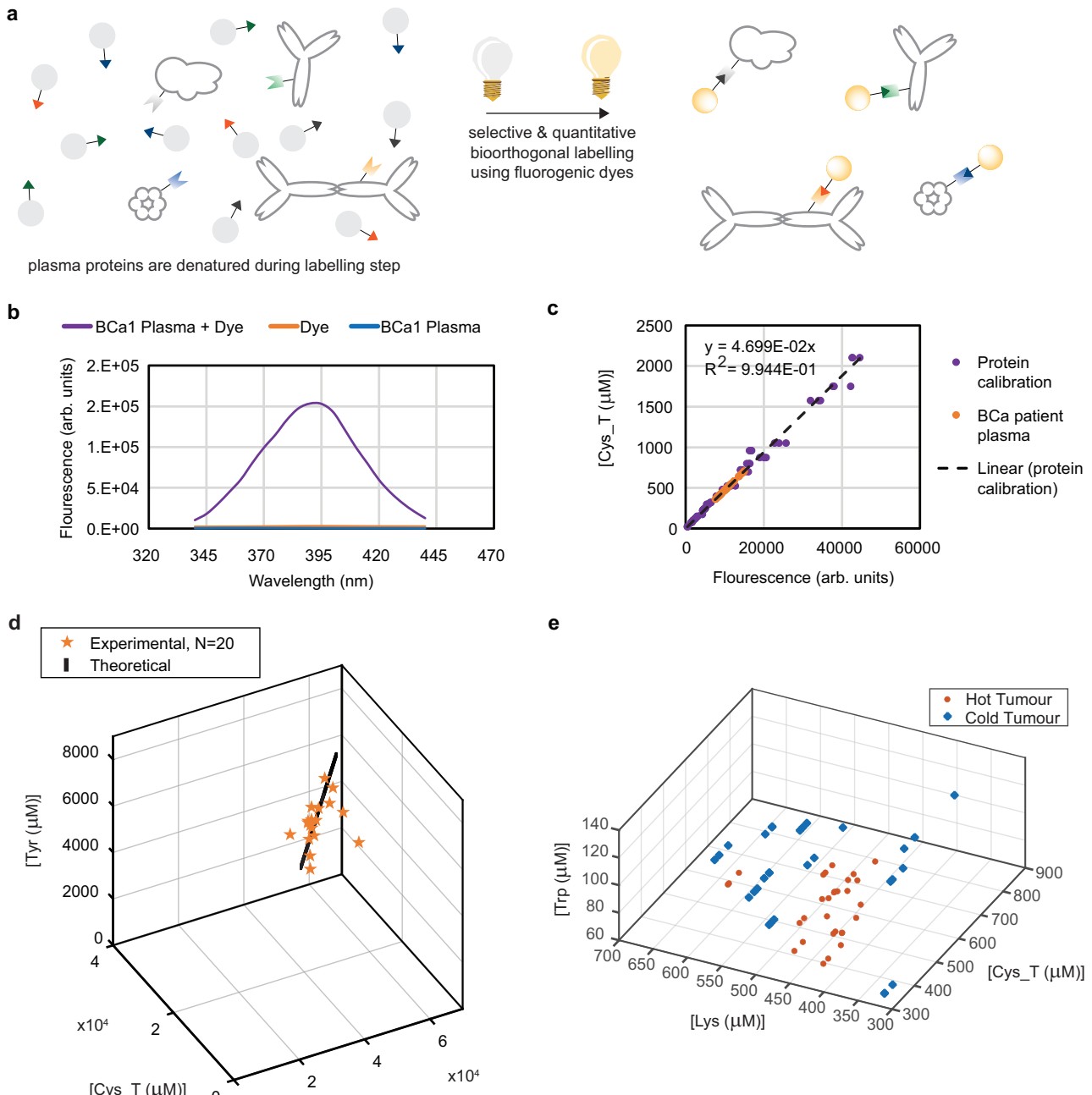

**Fig. 3 | Quantifying amino acids in neat patient plasma with bio orthogonal labelling. a** Our strategy involves performing five bioorthogonal labelling reactions. Each label is targeted to protein-incorporated amino acid residues of a specific amino acid type and reacts only with that amino acid type. Labels are fluorogenic, with fluorescence turning on exclusively after reaction with the targeted amino acid type of interest. Labelling is quantitative, so fluorescence intensity scales quantitatively with the concentration of the targeted amino acid type. **b** We identified fluorescence imaging regions where we did not observe autofluorescence from neat patient blood plasma. Fluorescence excitation spectra for labelling of protein-incorporated cysteine residues within the blood plasma of a patient with breast cancer, for the dye before mixing with the patient plasma, and for the patient plasma alone. **c** Because labelling is quantitative and no autofluorescence is observed from patient plasma, we use a calibration curve derived from solutions of known amino acid concentrations - Bovine Serum Albumin (BSA), Beta-Lactoglobulin (BetaLac), and Lysozyme (LYZ) of known protein sequence at known protein concentration − to determine the relationship between

fluorescence intensity and amino acid concentration for each amino acid type. We used the quantitative linear fit to the protein calibration curve to transform the fluorescence intensity measured for cysteine labelling in the patient plasma samples into the corresponding concentration of cysteine amino acid residues. Data shown for triplicates of $N = 20$ breast cancer patients plasma samples. **d** The relationship between the experimentally measured *Amino Acid Concentration Signature* (AACS) with theoretically calculated AACS from the known concentrations of individual plasma proteins. The AACS of Tyrosine (Tyr), Cysteine (Cys_T) and Lysine (Lys) are shown in the graph. Data shown for triplicates of $N = 20$ healthy patient plasma samples. **e** To investigate whether changes in the host immune response to tumour development drove AACS alternations, we measured the AACS of mice which had been injected with an immunologically Hot Tumour or a Cold Tumour. Note: Throughout, Cys_R refers to free cysteine residues that are not disulfide bonded, while Cys_T refers to total cysteine residues, including those involved in disulfide bonding.

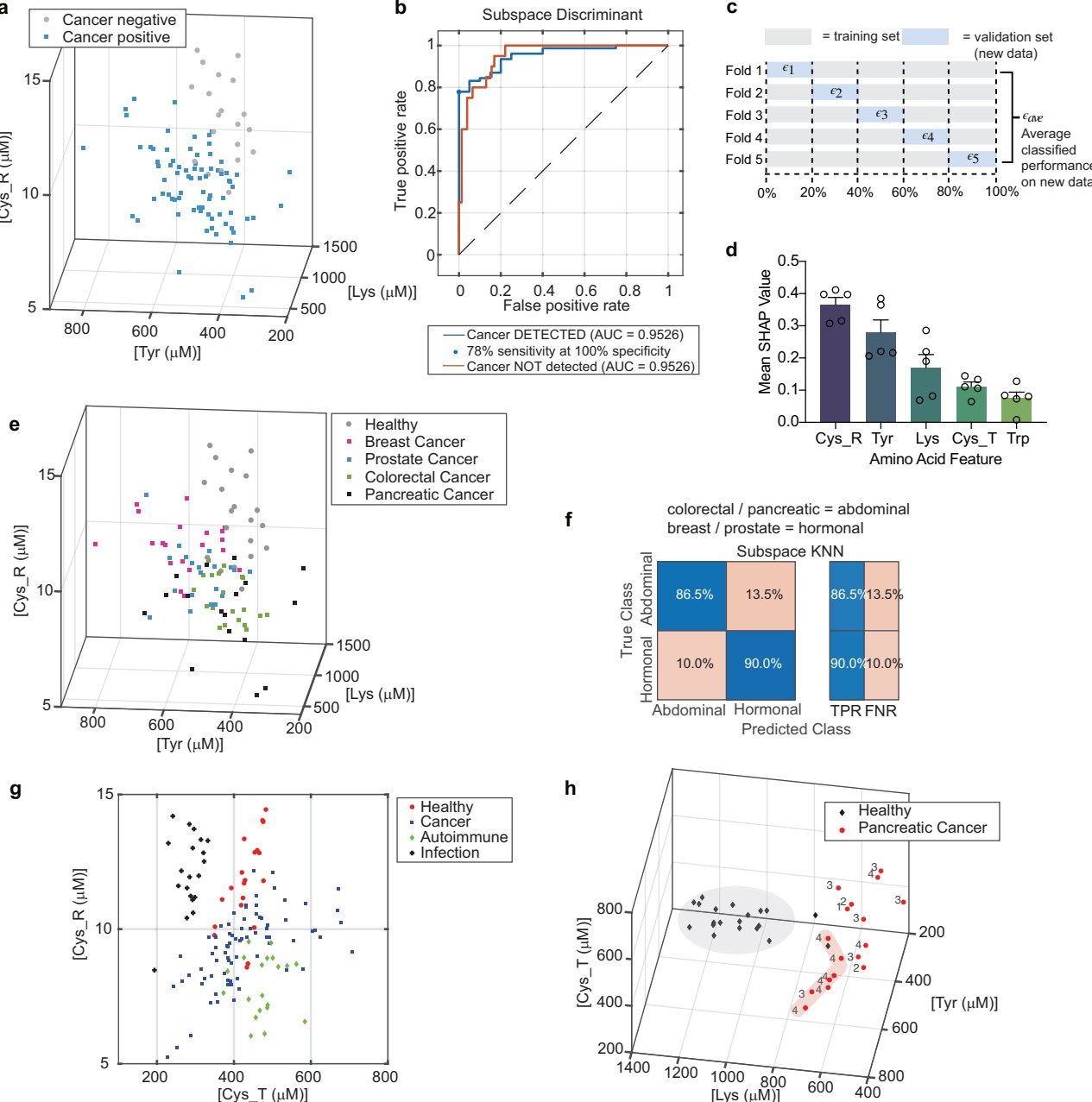

**Fig. 4 | Multi Cancer Early Detection using AACS. a** AACS signature measured from N = 77 cancer patients (blue squares) or N = 20 healthy donors (grey circles), plotted in N-dimensional space. 3/5 measured AACS dimensions shown for visual clarity, selected by choosing the 3 dimensions with the highest feature importance in the ANOVA analysis presented in Supplementary Fig. 5. **b** Receiver Operating Characteristic (ROC) Curve examining the accuracy of ensemble subspace discriminant classifier performance on unseen validation data. Area Under the ROC Curve was 0.95. **c** Cross-validation approach scheme. **d** Average normalised SHAP values with standard errors for a linear Support Vector Machine (SVM) model trained and validated on the N = 97 datapoints using 5X cross validation. **e** AACS measured and plotted in N-dimensional space, with patients labelled according to tumour location: breast (pink squares), prostate (blue squares), colorectal (green squares), pancreatic (black squares), and cancer-negative controls (grey circles). **f** K-nearest neighbour classifier performance on a held-back, unseen validation set trained and validated using cross-validation as in panel (**c**) to identify tumour origin. Cancers are localised to either abdominal (colorectal, pancreatic) or hormonal

(breast, prostate) origin. Abdominal cancers could be identified via abdominal CT scan, and hormonal cancers could be triangulated for further imaging or biopsy, considering patient biological sex. **g** AACS measured and plotted for N = 77 cancer patients (blue squares) and N = 20 cancer-negative controls (red circles), with the addition of AACS signatures for N = 20 patients with autoimmune disease (green diamond) and N = 20 patients with infection (black plus), providing controls of non-cancer heightened immune surveillance. 2/5 measured AACS dimensions shown for visual clarity and were selected using the ANOVA analysis in Supplementary Fig. 10. An Additional 3 dimensions provide required specificity, and UMAP representations are shown in Supplementary Fig. 8 and 9. **h** Healthy (black diamond) and pancreatic cancer (red circles) AACS, with the clinical stage of each pancreatic cancer patient shown numerically next to its data point. Patients with metastatic cancer cluster towards the healthy distribution, highlighted schematically with a shadow, whereas patients with earlier stages of cancer have signals further from the healthy distribution.

acid type using an ANOVA statistical test, as shown in Supplementary Fig. 5. The Cys_R, Lys, and Tyr dimensions had the largest contributions to the difference between the cancer-positive and cancer-negative distributions. We note that while ANOVA is conventionally used with linear models, in our analysis, ANOVA was applied not as a model inference tool, but to statistically assess the differentiation of individual amino acid levels between cancer and healthy groups.

Next, to determine whether we could predict whether an individual patient was cancer positive or cancer negative using our measured AACS plasma signature, we evaluated selected supervised machine learning approaches. Because we had established via a MANOVA that our cancer-positive and cancer-negative AACS data come from independent random samples from different normal distributions, we focused on discriminant analysis classification algorithms, which assume that different classes generate data based on different normal distributions.

We used K-fold cross-validation to ensure that the classifier generalises well to unseen validation data in the context of small to moderate sample sizes, up to several hundred patients in diagnostic applications. It protects against overfitting and ensuring that the classifier does not simply memorise characteristics of the training data. In this approach, the dataset is partitioned into 5 folds, and within each fold, 4/5 of the data is used for training and 1/5 of the data is unseen during training and is reserved for validation. The average of the three technical replicates per patient was used to ensure each partition contained only statistically independent biological replicates. Validation accuracy is estimated within each fold, and the results are averaged to provide a view of classifier accuracy when all the data is held back and used for validation. Therefore, we had an unseen validation set of $N = 97$ (Fig. 4c).

We evaluated the performance of linear discriminant, quadratic discriminant, and ensemble subspace discriminant classifiers. Ensemble subspace discriminant classifiers use weak learners to sample data from a subset of possible predictor dimensions, average the score prediction of the weak learners, and predict that samples belong to the category with the highest score. Here, we allowed 30 weak learners to sample 3 out of 5 possible (AACS) dimension predictors. We observed accuracies within the unseen validation set of 86.6, 87.6, and 89.7% respectively. Among these, we selected the ensemble subspace discriminant classifier because the Receiver Operating Characteristic (ROC) curve indicated that the classifier could operate at high sensitivity and very high specificity: 78% sensitivity at 100% specificity, using a decision threshold of 0.69 (Fig. 4b). Satisfyingly, this compares favourably to the sensitivity and specificity reported for ctDNA based liquid biopsy approaches, which at 99.5% specificity reach just 51.5% sensitivity[6].

To improve the interpretability of our classifier and assess robustness, we applied SHAP (SHapley Additive exPlanations) analysis to the linear SVM model. SHAP provides feature-level attributions, allowing us to interpret the contribution of each amino acid feature to the model's prediction. Using five-fold cross-validation, we computed normalised SHAP values for each fold and examined their variability. The ranking of feature importances was found to be statistically consistent across folds (Friedman $\chi^2 = 5.96$, $p = 0.201$), suggesting that the model interpretation is stable and unlikely to be influenced by overfitting. Figure 4d summarises the average normalised SHAP values with standard deviations, showing that Cys_R, Tyr and Lys contributed most to model predictions, which aligns with features identified via ANOVA analysis (Supplementary Fig. 5).

We further randomised the dataset into a $N = 73$ member training/validation set, and a $N = 24$ test set, and observed an AUROC in the held-out independent test set of 0.99 (Supplementary Fig. 6a, b). When staggering leave-out-one cross validation across our entire $N = 97$ set, we obtained comparable results as explained in Supplementary Fig. 7.

## Cancer specific immunosurveillance

The immune system activates not only in cancer but also in auto-immune conditions and infections[13–15]. Autoimmune conditions, and in particular connective tissue disorders such as rheumatoid arthritis, have been shown to exhibit the highest levels of non-cancer total IgG increases and immune activation compared to healthy patients.

Infection canonically activates infectious agent-specific immunoglobulins, rather than total antibody levels[27]. For example, SARS-CoV-2 infection significantly increases the SARS-CoV-2-specific IgG levels[28]. However, we did not want to rule out that infection could also change total antibody levels, or affect our biomarker concentrations via cascade effects from the acute phase reaction[29]. Therefore, we measured the plasma samples from $N = 20$ additional patients with an autoimmune disease and $N = 20$ patients with infection to determine whether the AACS immune response was cancer-specific or reflected general immune activation. This took our total cancer detection cohort to $N = 137$.

Gratifyingly, we observed in Fig. 4g and Supplementary Figs. 8 and 9 that the signals measured for patients with cancer, without cancer, with autoimmune conditions and infections were all different, verifying that our biomarkers measure a cancer-specific immune response. In addition, we performed a MANOVA analysis to test the null hypothesis that the cancer, healthy, autoimmune and infection distributions were the same, and obtained $p = 1.022 \times 10^{-10}$. Therefore, we reject the null hypothesis and conclude that each disease state has a distinct biomarker distribution at the 1% significance level. Furthermore, we investigated the importance of each amino acid type to the differentiation between healthy, cancer, autoimmune diseases and infections via an ANOVA as shown in Supplementary Fig. 10. We found that each of the 5 included amino acid types (Cys_T, Cys_R, Lys, Trp, and Tyr) had a statistically significant difference between each group, highlighting the possibility of increasing specificity against symptomatic controls via inclusion of additional AACS biomarkers.

Clinical and demographic characteristics of our entire cohort are provided in the Supplementary Information. Given that others have identified undulating changes in the human plasma proteome with age[30], we additionally explored whether age could be a confounding variable in cancer detection or enhance classifier performance. As shown in Supplementary Fig. 11, none of the AACS biomarkers were strongly correlated with age. When we trained the same classifier used in Fig. 4b to include age as an additional predictor, a comparable AUROC of 0.95 was achieved, but sensitivity at 100% specificity dropped from 78% to 61%, as shown in Supplementary Fig. 12. Collectively, the results suggest that age is not a significant confounding variable, and its inclusion would not improve cancer prediction performance.

## Identification of tumour site

The changing immunosurveillance patterns previously reported are specific for individual cancer types are cancer type specific[14]. Therefore, having established that we could accurately predict cancer presence using our AACS measurements, we next investigated whether, when cancer is predicted, we could additionally identify the cancer type to provide direction in the follow-up.

Our set of $N = 77$ cancer patients included 20 patients with breast cancer, 20 patients with colorectal cancer, 20 patients with prostate cancer, and 17 patients with pancreatic cancer. The AACS measurements of patients of each cancer type are shown in Fig. 4e. Gratifyingly, we observe differences in the distributions of pancreatic and colorectal cancer, and the gastrointestinal cancers are closer to each other in N-dimensional space than they are to the hormonally driven breast and prostate cancers which occupy similar positions in N-dimensional space. This is the goal of an embedding approach. All are different from the healthy distribution.

We then trained machine learning classifier to predict, among patients with a confirmed cancer prediction, the type of cancer and hence the tumour origin site to provide direction in the follow-up. Thinking forward to clinical workflow, we focused on among positive cancer predictions identification of abdominal cancers, which could be detected by abdominal computed tomography (CT) scan, or hormonally driven cancers (breast and prostate). Given the dependence of breast and prostate cancer on biological sex, these hormonally driven cancers could be confirmed via exam or imaging of the biological sex determined site. We trained an Ensemble Subspace Discriminant K-Nearest Neighbour classifier using 5-fold cross validation as shown in Fig. 4c. The results are shown in Fig. 4f. Among patients with a positive cancer prediction, we could identify the tumour location with 88% accuracy. This result is particularly interesting given that most of our cancer patients were metastatic, with similar sites of metastasis (see clinical information in Supplementary Data 1). These results suggest that primary tumours of different types elicit different immune surveillance responses.

**Stronger signals for early stage**

Conventional liquid biopsy ctDNA based approaches have struggled, particularly with the detection of early-stage tumours, some of which have been observed to release little to no ctDNA. In recent reports, only 8% of stage 1 tumours have been detected[5]. In general, the ctDNA signal measured scales with tumour burden, with tumours of greater size shedding more DNA associated with apoptosis into circulation. When we analysed AACS signals in comparison to cancer stage, we observed that satisfyingly, our AACS signals showed the opposite trend, increasing in strength (distance from the healthy distribution) for earlier-stages of pancreatic cancer (Fig. 4h). Immune activation is stronger for earlier stages of cancer before immune tolerance/evasion allows metastatic dissemination[10]. Therefore, we would expect AACS signals to be stronger for earlier stages of solid tumour cancers in general.

**Predicting response to first-line CDKi treatment in breast cancer patients**

Cyclin-dependent kinase inhibitors (Palbociclib, Ribociclib, and Abemaciclib) have been transforming the first-line management of patients with the most common subtype (hormone receptor positive, HER2-) of advanced breast cancer[31]. Abemaciclib has recently been approved for adjuvant treatment of early-stage breast cancer at high risk of recurrence[32]. Because they target a general mechanism of cancer cell progression through the cell cycle (Fig. 5a), this class of oral drugs is being tested for the treatment of solid tumour cancers in general and haematological malignancies. There are over 40 public new CDKi assets in pharmaceutical development pipelines[33].

However, as often occurs with anti-cancer therapies, questions have raised regarding the identification of breast cancer patients who do not benefit from CDKis treatment. Among the approved CDKi's, approximately 30% of baseline naïve patients have been clinically observed to have primary resistance to treatment meaning that they continue to progress on CT scan within 6 months of treatment initiation. Because 30% of non-responding patients have very low (< 6 months) Progression-Free Survival (PFS), treatment of non-responding patients obscures PFS benefit for the responding patients who usually have PFS > 24 months. The SONIA trial recently found that first-line treatment with CDKi's did not result in statistically or clinically meaningful PFS increases compared to second-line treatment, while doubling patient side effects and significantly increasing costs[34].

There is no biomarker of primary resistance to CDKi's. Genomic and proteomic assays examining the targets suggested by the canonical cell cycle mechanism have not yielded clinically meaningful predictions[35]. Efforts to use ctDNA levels to predict response in baseline naïve patients have also been unsuccessful[7]. Recently, an additional CDKi mechanism has been discovered (Fig. 5a). Independent of their role as cell cycle regulators, CDKi's suppress the production of

regulatory T cells. These cells create immune tolerance and are associated with decreasing immunosurveillance during cancer metastasis. CDKi's have been shown to increase immunosurveillance[36].

Because our platform detects changes in immunosurveillance, we hypothesised that we could predict response to CDKi's. To test our hypothesis, we measured the plasma AACS signatures from $N = 33$ hormone receptor positive, HER2 negative advanced breast cancer patients who were recommended for but had not yet started CDKi treatment, meaning that they were baseline naïve for CDKi's.

As described in Fig. 5b, patient whole blood samples were collected, and we measured 500 µL plasma from each patient in triplicate. We observed low Coefficients of Variation (CVs) across each labelling dimension. The patients included those with bone only, visceral only, and bone plus visceral metastatic spread. We investigated whether subsequent patient CDKi response – assessed by CT scan 6 months after starting treatment – explained the variation in the patients measured AACS plasma signatures. In Fig. 5c, we plot the results in N-dimensional space. Visually, the CDKi Responder and Non-Responder signatures appear distinct, with the CDKi Non-Responders clustering in a tight region of the graph.

To investigate whether subsequently assessed patient primary CDKi response statistically explained the variation in the measured AACS immunosurveillance data, we performed a Multidimensional Analysis of Variance (MANOVA) statistical test to test the null hypothesis that Non-Responding and Responding CDKi patients had the same set of AACS expression. We calculated the p-value using Wilks' Lambda, and found that patient response/non-response explained the variation in AACS at the 5% significance level ($p = 0.0461$).

Having established that the AACS immunosurveillance data for the patients who would go on to respond and not respond to CDKi's were different, we next explored using machine learning whether we could predict the response of an individual patient to CDKi's. To do this, we trained a linear Support Vector Machine (SVM) classifier using 5-fold cross validation as shown in Fig. 4c which validates classifier performance using unseen data. As shown in Fig. 5d, we evaluated the true positive and false negative rates of classifier predictions. We correctly predicted 100% of the majority class (responding patients), while still detecting 43% of the minority class of non-responding patients via their AACS immunosurveillance signatures. Because the sample class distribution was representative of the population distribution (22% non-response), we were also able to evaluate the positive predictive value and false discovery rate of classifier performance, as shown in Fig. 5e. When our approach predicts that a patient will not respond to CDKi's, this is correct 100% of the time, and when it predicts that a patient will respond, it is correct 87% of the time. Hence, it could be used to accurately stratify patients for CDKi treatment or trials.

Moreover, to assess the potential for overfitting and improve the interpretability of the linear SVM model used for predicting response to CDKi treatment, we performed SHAP analysis. We similarly conducted five-fold cross-validation with stratified sampling and calculated normalised SHAP values for each fold to evaluate the consistency of feature importance across training subsets. To statistically examine the stability of feature rankings across folds, we further applied the Friedman test, which yielded a non-significant result ($\chi^2 = 5.60$, $p = 0.231$). This indicates that the ranking of feature importances does not significantly vary between folds, supporting the conclusion that the model's interpretability is stable and not highly sensitive to the specific subset of training data used. Figure 5f summarizes the mean normalised SHAP values and standard deviations, illustrating which amino acid features consistently contribute to model predictions.

This cohort of patients were prospectively followed and the RECIST criteria was adopted to evaluate disease response, stabilisation or progression. Among our $N = 33$ CDKi patients, 10 had progressed by the time of writing. Their plasma sample was measured again as soon

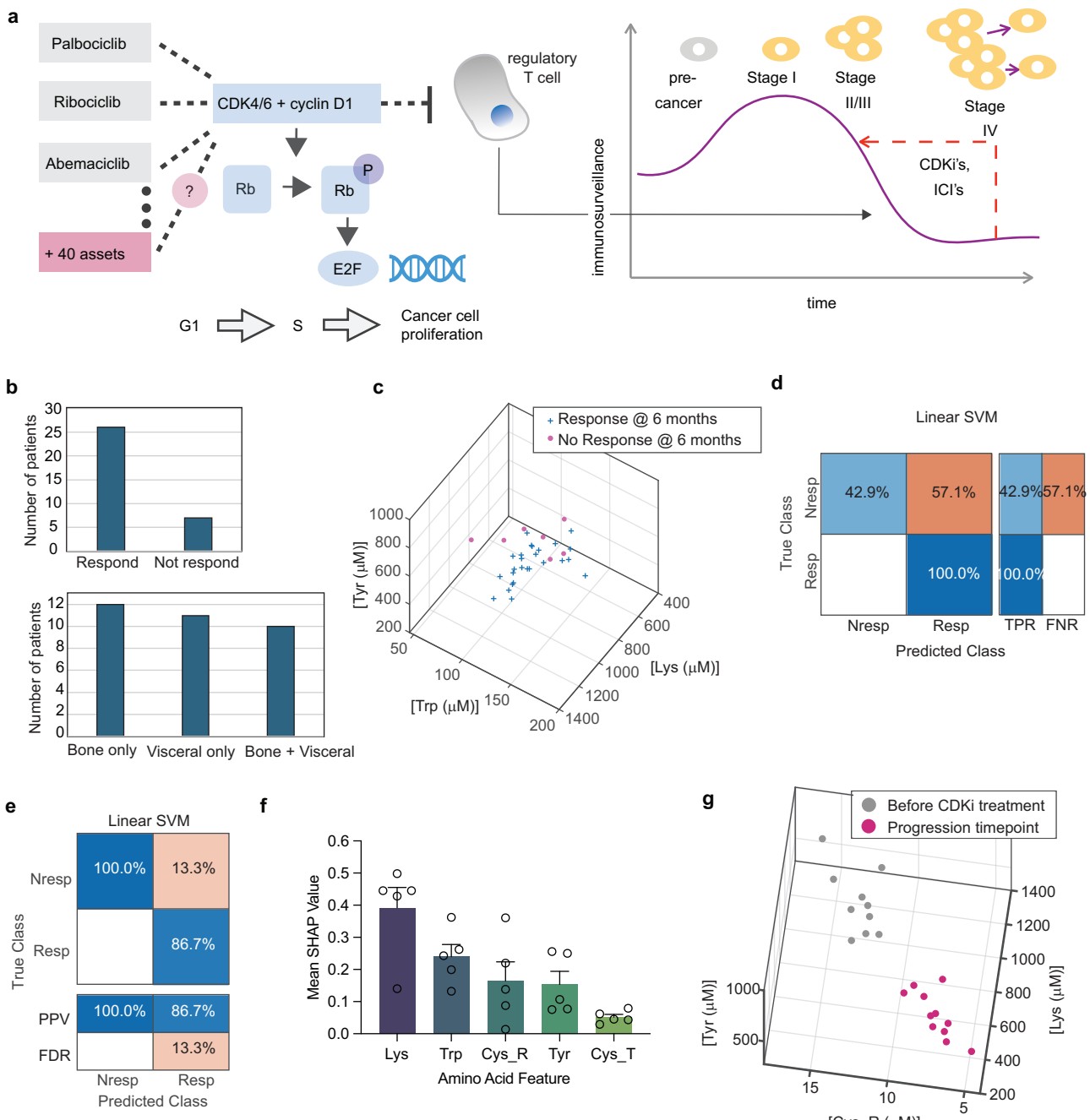

**Fig. 5 | Companion Diagnostic using AACS. a** Cyclin-dependent kinase inhibitors (CDKi) mechanisms of action[41,42]. Canonically, CDKi's (including Palbociclib, Ribociclib, Abemaciclib, and assets in development) inhibit cancer cell proliferation through the cell cycle by inhibiting CDK4/6 binding to cyclin D1. This limits phosphorylation of Rb, needed for activation of E2F-associated cell cycle control genes. In addition, CDKi's have recently been shown to suppress production of Regulatory T cells. These cells create immune tolerance and are associated with decreasing immunosurveillance during cancer metastasis. In this way, CDKi's increase immunosurveillance. **b** Clinical characteristics of *N* = 33 Hormone Receptor positive, HER2 negative metastatic breast cancer patients who were recommended for CDKi's according to current treatment guidelines. Patients were prescribed CDKi's as per standard of care, but before starting treatment, their blood sample was collected and analysed using AACS. Their response was evaluated on a 6-month CT scan according to RECIST criteria. Non-responding patients had cancer progression by 6 months. **c** Pre-treatment AACS measured for CDKi-prescribed patients plotted in N-dimensional space, coloured according to subsequently assessed response at 6 months. 3/5 dimensions shown for visual clarity. The 3 presented features were selected using an ANOVA feature ranking in Supplementary Fig. 13. **d** A linear SVM classifier was trained and validated using held-back, unseen validation data. True positive rate (sensitivity) and false negative rate (1 - specificity) on the validation data. This metric evaluates what percentage of true responders and non-responders are identified using the classifier. **e** Positive predictive value, and false discovery rate for the classifier described in (**d**). This metric evaluates when a prediction is made by the classifier, what percentage of the time is it correct. **f** Average normalised SHAP values with standard errors for a linear Support Vector Machine (SVM) model trained and validated on the *N* = 33 datapoints using 5X cross validation. **g** Among *N* = 33 CDKi-treated patients, 10 had progressed by the time of writing. Their plasma sample was measured again when progression was identified, and the AACS results compared in N-dimensional space for before they started CDKi treatment versus at the timepoints where their progression was identified.

as progression was identified, and the AACS results compared in N-dimensional space for before the started CDKi treatment versus at the timepoints where their progression was identified (Fig. 5g). This data suggests that there is a distinct signature of progression on CDKi treatment, measurable with AACS biomarkers.

## Discussion

Our strategy fuses chemistry, chemical biology, biology, mathematical, and machine learning approaches to create a class of diagnostic assay. Our approach conveniently reveals changes in immune surveillance via simple measurements of the total concentration of amino acid residues within patient plasma samples. We confirmed via a mouse model that this reveals primarily changes in the host immune response. Traditional isolation steps required prior to proteomic, genomic, or metabolic assays are eliminated in our elegant and scalable microplate reader assay. Importantly, our immune surveillance signatures, measured via AACS, are distinct from those measured from non-cancer immune activation due to infection or autoimmune diseases.

Within the area of precision medicine, thus far, we have established that we can predict response to the three currently approved CDKi's. We anticipate that our approach will have broad applicability in predicting response to the 40 additional CDKi assets in development. Where data is available, the use of existing drugs in new indications or additional development assets have struggled to achieve the response rates of the existing approvals[37]. Therefore, we anticipate that the co-development of companion diagnostics using our AACS platform may be used to enhance PFS and OS by identifying a group of patients who will benefit from treatment with the investigational drug, facilitating drug approval. We note that our approach could also find application in predicting response to other immunosurveillance modifying drugs such as Immune Checkpoint Inhibitors (ICI's), where response rates vary from 15-60% with the quest to identify predictive biomarkers a major focus of the oncology research community[38].

Because changes in immune surveillance are greater for earlier stages of cancer, as shown in Fig. 1b, our approach is well suited to detecting the earliest stages of malignant transformation and to MCED. We hypothesise that differences in immunosurveillance can contribute to the development of cancer, in a similar way to how differences in proteostasis[39] can contribute to the development of protein misfolding diseases. Therefore, biomarkers that can detect changes in immunosurveillance, like the approach disclosed herein, can have broad applicability both in detecting the earliest stages of cancer development, such as MCED, and in predicting response to treatment with immunosurveillance-modifying drugs. We note that the platform developed here is orthogonal to genetic approaches, such as those based on next-generation sequencing. It is also orthogonal to metabolomic approaches, such as GAGomes[39]. It could ultimately be combined with next-generation sequencing or metabolomic approaches for enhanced sensitivity and specificity in MCED, or to predict response to targeted oncology targets of interest.

In the present work we have validated our approach using a cohort of $N = 170$ measured clinical samples. Significantly larger scale clinical validation studies are now ongoing.

While Immunotherapeutics have become hallmarks of targeted oncology in recent years, our approach will open the area of immunodiagnostics and immunosurveillance to active and rapid diagnostic development. In doing so, we create a new toolset for precision medicine, orthogonal and complementary to genetic, proteomic, and metabolomic approaches.

Our toolset sheds light on central questions in oncology. A major question in cancer biology remains whether the immune system sees cancer cells as "self", like in an auto-immune disease, or "foreign" like in an infection[40]. Our cross-sectional AACS biomarker provides a quantifiable biochemical phenotype, wherein differences in n-dimensional amino acid space reveal differences in protein expression and phenotype. Because our cancer signatures are closer to autoimmune diseases in than to infection in n-dimensional amino acid space, our data suggests that the immune system regards cancer as "self".

## Methods

### Ethics statement

All research described in this study complies with relevant ethical regulations for research involving human participants and animal experimentation. All human samples were obtained with written informed consent, and the study protocol was approved by the Ethical Board of the Academic Medical Centre of Lisbon (CAML; CHLN/FMUL/IMM). The study complies with the principles of the WMA Declaration of Helsinki and the Belmont Report. All animal procedures were conducted at the Instituto de Medicina Molecular João Lobo Antunes (IMM, Lisbon) in strict accordance with Portuguese Law (Portaria 1005/92), the European Directive 86/609/EEC, and the Federation of European Laboratory Animal Science Associations (FELASA) guidelines. Experimental protocols were approved by the Portuguese veterinary authority – Direção Geral de Alimentação e Veterinária (DGAV) – and by the IMM Animal Ethics Committee (authorisation AWB_2021_03_GB_Targ CancerDrugs). Mice were inoculated subcutaneously with tumour cells and monitored for tumour growth. The maximum tumour burden permitted by the ethics committee was 1000 mm³, and this limit was not exceeded. Animals were euthanised by isoflurane overdose once this threshold was reached. Throughout the study, animals were regularly monitored, and no signs of suffering or distress were observed.

### Patients and Healthy donor blood samples collection

Samples were requested from Biobanco-IMM, Lisbon Academic Medical Centre, Lisbon, Portugal. 5 mL of peripheral whole blood samples from were collected from $N = 20$ volunteer healthy donors, $N = 20$ rheumatoid arthritis patients, $N = 20$ SARS-COV-2 patients, or $N = 77$ cancer patients followed at CHULN-Hospital Santa Maria, Lisbon at baseline (before starting new line of therapy) according to the Oncology Department guidelines for disease evaluation. The samples collected from cancer patients include: $N = 20$ colorectal cancer, $N = 20$ prostate cancer, $N = 20$ breast cancer, $N = 17$ pancreatic cancer, $N = 33$ hormone receptor positive / HER2 negative advanced breast cancer patients (with the indication for CDKi treatment). From the last, followed collection at different time points during the treatment or at the time when they progressed. All samples were collected with informed consent and approved by the Ethical Board of the Academic Medical Centre of Lisbon CAML (CHLN/FMUL/IMM). Experiments are conformed to the principles set out in the WMA Declaration of Helsinki and the Department of Health and Human Services Belmont Report.

### In vivo mouse study

Mouse colon adenocarcinoma MC38 cell was purchased from Kerafast (ENH204-FP). Mouse melanoma cell line B16F10 was a gift from Dr Karine Marie Serre and Dr Susana Constantino Santos from iMM. The cells were cultured in Dulbecco's Modified Eagle Medium (DMEM, Gibco, Thermo Scientific, #21969035), supplemented with 10% heat-inactivated foetal bovine serum (FBS) (Gibco, Thermo Scientific), 1x GlutaMAX (Gibco, Thermo Scientific), 1x penicillin/streptomycin solution (Gibco, Thermo Scientific). The cell was cultured at 37 °C in humidified conditions with 5% $CO_2$. All cell lines were authenticated by STR profiling and routinely tested for mycoplasma contamination using MycoAlert Mycoplasma Detection kit (Lonza, LT07-710). No contamination was detected during the study period. All animal experiments were conducted at the Instituto de Medicina Molecular João Lobo Antunes (IMM, Lisbon). Animal work was performed in strict accordance with Portuguese Law (Portaria 1005/92) and the European Guideline 86/609/EEC, and follow the Federation of European Laboratory Animal Science Associations guidelines and recommendations

concerning laboratory animal welfare. All animal experiments were approved by the Portuguese official veterinary department for welfare licensing – Direção Geral de Alimentação e Veterinária (DGAV) and the IMM Animal Ethics Committee (authorization AWB_2021_03_GB_Targ CancerDrugs). 8-week-old female and male C57BL/6 J mice (purchased from Charles River) were used in this study, with $1 \times 10^6$ MC38 or B16F10 cells inoculated subcutaneously in the abdomen of mice. Tumour growth was monitored over time, by performing bilateral vernier caliper measurements and mean tumour volumes were calculated using the formula (length x width$^2$) /2. Blood was collected from the cheek of mice on Days 5, 8 and 11 after tumour inoculation. Animals were observed regularly; tumours were measured as described before, and mouse weight was evaluated throughout the study. Mice were monitored regularly, and animals were euthanized by isoflurane overdose once tumour volume reached 1000 mm$^3$, in accordance with the institutional animal care guidelines. The maximal tumour size/burden permitted by the ethics committee was 1000 mm$^3$, and this limit was not exceeded during the study. No signs of animal suffering or discomfort were obseved, including no weight loss. The light/dark cycle was 14 h light/ 10 h dark (lights on at 07:00; lights off at 21:00). The temperature was 20–24 °C, and the relative humidity was 55 ± 10%, with controlled supply of High Efficiency-Particulate Air (HEPA) filtered air provided to individually ventilated cages. The maximum number of animals per cage was 5. Social isolation was avoided whenever possible. The type of food was autoclaved diet pellets RM3A (P), from SDS Special Diets Services (Product code: 801030). Food was placed in a grid inside the cage and provided ad libitum to the animals. The type of water was sterile water treated by reverse osmosis. Water was provided ad libitum to animals through bottles with a capillary hole.

## Plasma samples preparation

The plasma isolation was done within 4 h after collection, at the Institute of Molecular Medicine (IMM). Bardelli's protocol was used to obtain plasma samples from the whole blood samples. Briefly, the whole blood samples were collected within EDTA-coated tubes (Sarstedt, Cat# 02.1066.001) and were centrifuged at room temperature for 10 min at 1750 g (soft-without brake), then the supernatant (plasma) was transferred to fresh 15 ml falcon tube (Sigma-Aldrich, Cat# CLS430766), followed by another centrifugation at room temperature for 10 min at 3000 g. Lastly, around 500 µL of the plasma from each sample were obtained and transferred to Eppendorf carefully without disturbing the pellet, and stored at – 80 °C. To minimise freeze-thaw cycles, we aliquoted the 500 µL of each sample into six 80 µL aliquots in low protein binding Eppendorf tubes (Sigma-Aldrich, Cat# EP0030108302). A small number (e.g., 10–15) of patient samples were removed from the – 80 °C ultrafreezer every time and thaw them at room temperature until completely thawed. Mix the thawed samples via gentle pipetting carefully to avoid create bubbles and to homogenise any concentration gradients. Flash frozen was performed via submersion in liquid nitrogen for all aliquots and stored at – 80 °C until use. Thaw the aliquot at room temperature until it is completely thawed, followed by gentle pipetting to mix when the aliquot was needed for the experiment.

## Preparation of labelling solutions

Lysine (Lys) amino acid labelling buffer. Composed of 12 mM OPA (phthaldialdehyde, Sigma-Aldrich, Cat# P1378), 18 mM BME (2-mercaptoethanol, Sigma-Aldrich, Cat# M6250) and 4% w/v SDS (sodium dodecyl sulfate, Sigma-Aldrich, Cat# 75746) in 200 mM carbonate buffer, pH 10.5. Generally, 80.5 mg OPA was weighted out and 20 mL of 500 mM carbonate buffer, pH 10.5, was added followed by 63 µL BME. 20 mL H$_2$O and 10 mL of 20% w/v SDS stock solution was added. The solution was protected from light and heated to 65 °C for 10 min. The K labelling solution was allowed to cool to room temperature prior to filtering through a 0.45 µm syringe filter to remove any large particles.

Tryptophan (Trp) and Tyrosine (Tyr) amino acids labelling buffer. Composed of 0.2 M TCE (2,2,2-trichloroethanol, Sigma-Aldrich, Cat# T54801), 10 mM TCEP (Tris(2-carboxyethyl) phosphine hydrochloride, Sigma-Aldrich, Cat# C4706), 4% w/v SDS in 5 mM HEPES (Sigma-Aldrich, Cat# H3375), pH 7. Generally, 500 µL 1 M TCEP-HCl stock solution was added to 38.5 mL 5 mM HEPES buffer, followed by 964 µL TCE and 10 mL 20% w/v SDS stock solution.

Cysteine (Cys_T) amino acid labelling buffer. Composed of 5 mM ABD-F (7-fluoro-2,1,3-benzoxadiazole-4-sulfonamide, Sigma-Aldrich, Cat# F3639), 4% w/v SDS in 160 mM pH 10.5 carbonate buffer. Generally, 1 mL 200 mM carbonate buffer, pH 10.5, was added to 10 mg of ABD-F. The vial was closed and vortexed for a few minutes so that all the ABD-F dissolved. The ABD-F containing solution was then combined with 6.36 mL 200 mM carbonate buffer, and the ABD-F vial was washed with the carbonate buffer to ensure all the ABD-F solution was combined. Then, 1.84 mL 20% w/v SDS was added.

Free cysteine (Cys_R) labelling buffer. Composed of 20 mM TCEP in 5 mM HEPES pH 7. Generally, 1 mL 1 M TCEP stock was added to 49 mL 5 mM HEPES. The TCEP reduction solution is stable for around a week in HEPES buffer at 5 °C (< 20% oxidised in 3 weeks).

All the solution was then aliquoted into five aliquots and were sealed with parafilm and protected from light prior to storage at – 20 °C.

## Calibration plates preparation and measurement

A series of dilutions (in PBS) of standard protein samples were prepared to measure each labelling reaction as calibration plates. 5–60 µM BSA (Bovine Serum Albumin, heat shock fraction, Sigma-Aldrich, Cat# A7030) or Beta-Lac (Beta-Lactoglobulin from bovine milk, Sigma-Aldrich, Cat# L0130), 10-120 µM LYZ (Hen egg white Lysozyme, Sigma-Aldrich, Cat# 10837059001) and 10 µM PH (parathyroid hormone human, Sigma-Aldrich, Cat# P7036) were used as standard protein samples. The concentration of the standard protein samples was determined by NanoDrop 2000 Spectrophotometers (ThermoFisher Scientific, Cat# ND-2000) (mean of 3 replicates), considering the extinction coefficient (ε) (absorbance at 280 nm).

A 96-well plate layout was used for calibration plates for all labelling solutions (Supplementary Table 2).

The preparation and measurement of each labelling calibration plate are summarised in Supplementary Table 3. 96 well UV-Transparent microplate (Corning, Cat# CLS3635) was used for Trp/ Tyr/Lys labelling reactions due to the low excitation wavelengths used and required for UV transmission, 96-well NBS microplate (Corning, CLS# 3651) was used for Cys_T/Cys_R labelling reactions. The fluorescence measurements were performed using a Tecan Infinite 200 Pro microplate reader and a Tecan Spark multimode microplate reader with the indicated settings as mentioned below in the table.

## Patient samples measurement

The aliquot samples were diluted 1:60 in PBS for Trp/Tyr/Lys/Cys_T labelling measurements, diluted 1:6 for Cys_R labelling measurements, and 1:120 in PBS for protein concentration quantification. A 96-well plate layout was used for the patient samples for every labelling reaction (Supplementary Table 4). The preparation and measurement of each labelling reaction plate are summarized in Supplementary Table 5. The 96 well plates and microplate readers were used the same as described for Calibration plate measurement.

## Protein concentration quantification

The BCA assay (Bradford Colorimetric Protein Assay) was used to confirm the protein concentration determined via a simultaneous disease identification, following manufacturer's instructions. BSA protein standard was supplied by the manufacturer (1000 µg/mL stock concentration).

## Mathematical embedding

We realised that because proteins have common amino acid building blocks, the complex protein composition of plasma is well suited to an embedding, with each axis in n-dimensional space also having physical and biological relevance as revealing the total concentration of an amino acid building block.

We illustrate this concept in Fig. 2b, c. In Fig. 2c, we show the high-dimensional dataset of the plasma proteome in n-dimensional space (5-dimensional space corresponding to total concentrations of five amino acid residue types). Each row (protein) within the dataset is represented by a vector whose direction provides its number of amino acids of each of the five considered amino acid types, and whose magnitude is given by the protein concentration, as shown in Equation (1).

$$\mathbf{v} = \langle a_1\theta, a_2\theta, a_3\theta, a_4\theta, a_5\theta \rangle \tag{1}$$

Where $a_n$ denotes the number of the nth amino acid type present in the protein sequence and $\theta$ is the protein concentration.

In the embedded representation, as shown in Fig. 2c and Equation (2),

$$\mathbf{E} = \sum_{i=1}^{\alpha} \langle a_{1,i}\beta_i, a_{2,i}\beta_i, a_{3,i}\beta_i, a_{4,i}\beta_i, a_{5,i}\beta_i \rangle \tag{2}$$

wherein

$$\beta_i = \frac{\theta_i}{\sum_{i=1}^{\alpha}\theta}$$

The embedding, $\mathbf{E}$, determines a single vector of amino acid concentration values across all $i = 1$ to $\alpha$ plasma proteins by performing a linear combination of their $a_n$ values with their proportional composition within the plasma protein mixture, $\beta_i$. Therefore, every individual is represented by a unique datapoint within the n-dimensional space.

## Machine learning analyses

The linear discriminant, quadratic discriminant, ensemble subspace discriminant, and weighted K-nearest neighbour classifiers were performed and evaluated. The ensemble subspace discriminant classifier operates by leveraging weak learners to sample data from a subset of available predictor dimensions, average the score predictions generated by these weak learners, and classify samples into the category that receives the highest score. Here, we employed 30 weak learners to sample 3 out of 5 possible AACS dimension predictors. K-fold cross-validation was used to ensure that our classifier would generalise well to unseen validation data, especially given the challenge posed by small to moderate sample sizes.

## Statistics & reproducibility

All statistical analyses were performed using MATLAB (R2023a) or the Python programming language (version 3.10.9). MANOVA methods were conducted using the 'MANOVA' function from the statsmodels package. No statistical method was used to predetermine sample size. All collected data were included in the analyses, and no data were excluded. Randomisation and blinding were not applied, as they were not applicable to the design and objectives of this study. Detailed sample sizes and statistical test results are provided in the figure legends and relevant sections of the manuscript.

## Reporting summary

Further information on research design is available in the Nature Portfolio Reporting Summary linked to this article.

## Data availability

All data supporting the findings of this study are available within the article. The clinical and demographic characteristics of all patients involved in the study are provided in the Supplementary Data 1. No additional raw or processed datasets require deposition in external repositories. Source data underlying graphs and charts in the main figures are provided as a Source data file. Source data file is provided with this article. Source data are provided in this paper.

## Code availability

The machine learning models and mathematical embeddings generated and analysed in this study are proprietary and associated with ongoing development by a startup company. As such, the code and related resources are not publicly available. However, in accordance with Nature Communications policies, they are available to qualified academic researchers for non-commercial use upon request. To request access, please contact the corresponding authors, G.J.L.B. (gb453@cam.ac.uk) and E.V.Y. (emma@proteotype.com). Requests will be reviewed and responded to within 14 days. Access, if granted, will be provided for academic use only and may be subject to a material transfer or data use agreement.

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

## Acknowledgements

This project has received funding from Proteotype Ltd. and Fundação para a Ciência e a Tecnologia (2022.08101.CEECIND to C.T., UIDB/00124/2020, UIDP/00124/2020 to Q.H., Social Sciences DataLab - PINFRA/22209/2016 to Q.H.).We also thank Biobanco-GIMM, Lisbon Academic Medical Centre, Lisbon, Portugal (Ângela Afonso, Andreia Lopes, Ionela Toader and José António Cordeiro Torres Maximino) for processing, preparing and storing patient samples.

## Author contributions

C.T., L.C., E.V.Y. and G.J.L.B. conceived the study. E.V.Y. and G.J.L.B. supervised the study. C.T. performed microplate reader assays. P.C. and A.C. prepared plasma samples from breast, prostate and colorectal cancer patients. P.C., A.L.B. and S.C. reviewed clinical information. C.O.M. and J.E.F. provided rheumatoid arthritis patient samples and related clinical information. C.M.A., C.A., G.N.C., L.R., R.S. and L.C. recruited CDKi cohort patients and provided the samples. E.V.Y., Q.S., Q.H., S.W., and W.S. performed machine learning and statistical analysis. C.T., E.V.Y. and G.J.L.B. wrote the manuscript with contributions from all authors.

## Competing interests

W.S., E.V.Y. and G.J.L.B. are co-founders of Proteotype Diagnostics Ltd. C.T., W.S., L.C., E.V.Y. and G.J.L.B. are stockholders of Proteotype Diagnostics Ltd. W.S. and E.V.Y. are employed by Proteotype Diagnostics Ltd. Proteotype Diagnostics Ltd owns a patent application that incorporates a method of identifying the presence and/or concentration and/or amount of proteins or proteomes, which is described in this manuscript (EP4196797A1). G.J.L.B. is a Visiting Professor at Xi'an Fengcheng Hospital. All other authors declare no conflict of interest.

## Additional information

[1]GIMM - Gulbenkian Institute for Molecular Medicine; Avenida Prof. Egas Moniz, Lisboa, Portugal. [2]Proteotype Diagnostics Ltd, Babraham Research Campus, Cambridge, UK. [3]Xi'an Fengcheng Hospital, No.9 Fengcheng Third Road, Xi'an, Shaanxi, China. [4]Nova School of Business and Economics, R. da Holanda 1, Carcavelos, Portugal. [5]Serviço de Oncologia Médica, ULSSM, Unidade Local de Saúde de Santa Maria, Lisboa, Portugal. [6]Serviço de Reumatologia, ULS de Santa Maria, Centro Académico de Medicina de Lisboa, Lisbon, Portugal. [7]Faculdade de Medicina, Universidade de Lisboa, Centro Académico de Medicina de Lisboa, Lisbon, Portugal. [8]Yusuf Hamied Department of Chemistry, University of Cambridge; Lensfield Road, Cambridge, UK. [9]Translational Chemical Biology Group, Spanish National Cancer Research Centre (CNIO), Madrid, Spain. ✉e-mail: cong.tang@gimm.pt; luis.costa@ulssm.min-saude.pt; emma@proteotype.com; gb453@cam.ac.uk

