## [Transparent Peer Review file · Nature Communications]

Immunodiagnostic plasma amino acid residue biomarkers detect cancer early and predict treatment response

Corresponding Author: Professor Gonçalo Bernardes

Version 0:

Reviewer comments:

Reviewer #1

(Remarks to the Author)

General Comments:

In this paper, the authors have developed a novel immunodiagnostic platform to detect and identify a panel of amino acid residue biomarkers that provide a signature of cancer-specific immune activation associated with tumor development and distinct from autoimmune and infectious diseases, while being conveniently measured optically in neat patient blood plasma.

By simply measuring the total concentrations of cysteine, free cysteine, lysine, tryptophan, and tyrosine protein-incorporated biomarkers and analyzing the results with supervised machine learning, they can identify 86% of cancers with almost 0% false positive rate (N=92).

They also demonstrated that the cancer, healthy, and autoimmune/infectious biomarker pattern changes are all statistically significant with p value smaller than 0.0001. In their study the smaller scale changes in biomarker concentrations reveal inter-patient differences in immune activation which predict response to treatment. Specific concentration ranges of cysteine, free cysteine, lysine, tryptophan, and tyrosine biomarkers are shown to be predictive of response to Cyclin- dependent kinase inhibitors (CDKi's) in advanced breast cancer patients (p-value smaller than 0.05) and allowed identification of 100% of responding patients (N = 30). Their findings highlight the potential of the systemic immune response in cancer diagnostics. Furthermore, differences in amino acid biomarker space provide a cross sectional view of the biochemical phenotype differences between disease states providing new insights into central questions in cancer biology.

Specific comments (Statistical):

1. The use of Machine Learning techniques to identify the biomarkers is becoming common these days but identification of immunodiagnostic biomarkers indeed is very novel in this paper.
2. Since this is a high dimensional problem, the designing the experiment is very tricky. But the authors of successfully designed the experiment by using a novel biological embedding technique.
3. The authors have performed a very sophisticated data analysis to achieve their goals.
4. MANOVA to detect multi cancer early detection is very effectively done for clustering the types.
5. The classification using cross-validation sample is also done very effectively.
6. Overall, the results are also very significant and the study is very informative, novel and the state-of-the art.

Reviewer #2

(Remarks to the Author)

Title: Immunodiagnostic plasma amino acid residue biomarkers detect cancer early and predict treatment response
In the current study, Tang and colleagues established a panel of amino acid residue biomarkers for cancer-specific immune activation that is associated with tumor development and that is distinct from autoimmune and infection disease. By measuring total concentrations of cysteine, free cysteine, lysine, 38 tryptophan, and tyrosine protein-incorporated biomarkers, coupled with supervised machine learning, authors were able to identify 86% of cancers (N=92) at 100% specificity. Smaller scale changes in biomarker concentrations showed inter-patient differences in immune activation that predicted response to CDKis in the context of breast cancer.

Overall, the study is of considerable interest. However, there are several concerns.

1- Subject characteristics. A table describing subject characteristics among the comparative groups would be more easily interpretable compared to the excel file that has several tabs. There are also issues with reported DOB (for instance, healthy control #17 has a DOB of 31/01/1957 with a collection date of 20/06/2017. Assuming these were typos, age distributions among cases and controls appear quite different. Is there an association between reports biomarker values and patient characteristics, such as age and sex? Additionally, were patients fasted when blood was collected? Such information is important to assess for potential confounders.

2- Cancer staging- based on the supplemental table, it appears that most cancers are advanced stage (for instance, for PDAC all but 1 are stage III-IV). Thus, utility of the test multi-cancer 'early' detection is unclear and not supported by provided data.

3- While this reviewer appreciates the innovative approach, the cohort size is too small to make any definitive conclusions regarding utility of the test for MCED or for predicting response to treatment.

4- Although authors include patients with autoimmune and infection, other benign conditions pertinent to the field of cancer diagnosis were not included (e.g. benign polyps or low-grade dysplasia cysts, chronic pancreatitis, etc), precluding evaluation of specificity towards cancer. However, this is not a major demerit given challenges associated with acquisition of such specimens.

5- Classifier performance. Although authors used a validation set, these were based on the same cohort. As such, any potential confounder would be present in both the training and validation set. Additionally, see comment 2 above.

6- Figure 5- the number of patients that did not respond is very low (n= 5). Of note, the supplemental table indicates 4 non-responders rather than the 5 that are depicted in Figure 5C. Moreover, it does not appear that there is a great degree of separation, except for one individual. As mentioned above, the sample size is too small and subject to overfitting from models. Additionally, these analyses do not factor in patient and tumor characteristics (e.g. stage) that may be associated response rates.

7- Minor: Pancreatic cancer is not a prevalent cancer.

Reviewer #3

(Remarks to the Author)

In this study, the authors demonstrate a method to detect cancer at an early stage by measuring amino acid residues in plasma as biomarkers. They have developed a simple, useable workflow, leading to accurate predictions, which is striking. The use of the method to stratify patient groups is a particular highlight. However, I have a few questions about the method, the strength of some of the data, and its utility in particular situations.

1. It is not clear exactly how the authors settled on the final five amino acid residues to be tested. In Figure 2, they describe the composition of plasma, but in 2c, they do not consider the contribution of "c. 3000 other proteins". Were these proteins excluded from the search for target residues, in order to focus more on immunoglobulins as a readout of immune changes?
 - a. How the numbers of amino acid residues change in these other proteins should be shown.
 - b. How did the authors decide what constituted a substantial change in the number of amino acid residues? For example, reduced cysteine varies in number from 0 to 1, and disulfide cysteine is generally in the range 20 – 35 (other than 80 in IgM). What is the normal variability for the number of each amino acid residue in this set of proteins?
2. While the authors discuss that they want to examine amino acid residues in proteins, rather than free amino acid metabolites, it is not clear that they have taken any steps to address this in their biological processing. The plasma proteins are denatured during the labelling step (Fig. 3a), so how are free amino acids in plasma excluded from labelling?
3. It is not clear what point the authors are trying to make in Extended Data 3 – the amino acids shown change similarly in the different conditions, while the authors seem to suggest that the signals are different.
4. Fig. 4g needs more data points to be convincing.
5. It would be great if the authors had a particular piece of biological evidence pointing to the immune system being the main mediator of these effects. For example, does this method fail in patients that are already immunocompromised due to disease or drug treatment, and produce less immunoglobulin?

Reviewer #4

(Remarks to the Author)

Manuscript by Tang et. al is about a diagnosis method for cancer and prediction of treatment response based on plasma amino acid residue in terms of immunosurveillance. Authors set up an optical detection with fluorogenic labels for immunoglobulin components considering 5 amino acid residues. They applied this method to detect multi Cancer and introduced a discriminant classifier for prediction, as well as an KNN classifier for cancer location prediction. Additionally, they applied this method to evaluate the CDKi cancer treatment considering the difference of sensitivity of patients. The

major advantage of this method is the better sensitivity for early diagnosis compared with ctDNA methods. Author should address the following concerns for publication purposes.

The first issue is about the selection of embedded amino acids. Out of 20 major amino acids, authors selected 5 types based on biological embedding. However, the rationale to select those amino acids is not clear in figure 2c and the authors claimed "We identified N=5 amino acid types whose numbers changed substantially across the proportion-weighted fractions, such that they would be detectable in an average biological embedding signature." Author should also provide the information of other amino acids to show a comprehensive selection. In the method part, authors introduced an embedding formula as amino acid number weighted protein concentration. How many proteins are considered for this calculation? Which plasma proteome database is used?

As for the fluorogenic labels, authors considered confounding issues. It seems authors performed a positive/negative control in figure 3b and 3c to show the amino acid type is only related to the autofluorescence. Since the dye is only reactive to certain amino acids, I wonder if the authors also checked the calibration curve of amino acid with fluorescence and added this curve to figure 3c. Their slope should be similar and if not, the authors should consider matrix effects for this method.

Authors selected MANOVA to show the distribution between case and control and then employed machine learning algorithms for prediction. A regular machine learning or statistical learning process should use dimension reduction methods such as PCA and UMAP to show the separation of samples. However, this work always uses a 3D visualization with amino acids to show the separation, which is not clear in figure 4f and 4d. If the amino acids are selected appropriately, I would expect a clear separation on PCA or UMAP. Though a 5 fold cross validation is performed for training, the machine learning algorithms are not evaluated for real test data, which should not be used for training process and I will concern the overfitting issues. The machine learning algorithms for prediction of cancer location and type are different and I can't find the model selection process. I would suggest the authors use automatic machine learning tools such as autogluon for this dataset and the embedding of different models might further improve the results. The interpretable machine learning module of such tools will also tell the importance of different amino acids for different applications. Those results might give hints on the mechanism of immune surveillance for cancer.

Another important issue is about the discussion of biomarkers. It seems the core of this method is the selection of the 5 amino acids types and the authors didn't discuss why they can work. The overall hypothesis is about the immunology response of cancer and the reader might still doubt why 5 amino acid types are good and enough to stand for immunosurveillance. Meanwhile, those amino acids should also be covered by proteomics and metabolomics analysis and authors should cross check their results for support or disparity in their results. I will also suggest the authors try all of the 20 amino acid labeling to build a comprehensive residue model for cancer markers and the residual labeling technique might also be used for proteomics studies.

The last concern is about the figures. Some figures are not necessary to provide information such as figure 1d, 2a, and 4c. Those information are either hard to read or background knowledge from the textbook. Meanwhile, the 3D scatter plot is hard to read for cluster separation.

Version 1:

Reviewer comments:

Reviewer #1

(Remarks to the Author)

Early detection of cancer biomarker to predict treatment response.
The revision is satisfactory in terms of statistical contents.

Reviewer #2

(Remarks to the Author)

The authors were largely responsive to prior comments. Nevertheless, there are a few additional requests.

1) Please include a table that summarized patient characteristics described in individual tabs of the supplemental materials.
2) This Reviewer appreciates sub-analyses showing association between measured AAs and age (stratified by <50 and >= 50). However, these plots are based on a log10 scale. It would be of benefit to show correlation matrices between AA levels and age as continuous variables for cases, controls and the entire cohort. For classifier performance, it would be of benefit to include age as a co-variable.

Reviewer #3

(Remarks to the Author)

The authors thoughtfully addressed my comments and both clarified and strengthened key points of the paper. I recommend this work for publication.

Reviewer #4

(Remarks to the Author)

The authors have addressed my concerns on selection of amino acid type.

My second concern is not responded. I would like to know if pure amino acids instead of residual of protein can be labeled and show the same calibration curve. This analysis will get rid of the specificity issue of this methods when the calibration curve of amino acid show the same slope as protein.

The third concern has been addressed by UMAP. However, I found 3D figures in this paper showed different x, y, z axis.

There are 10 combinations to select 3 axis from 5 amino acids types and 3D figure could only show 3 amino acids. If the purpose for this visualization is show the separation of the amino acid selection, 3D figures will not get rid of the concern of cherry-picking. Meanwhile, the new figure 4g looks confusing. What's the meaning of the shadow? Why some dot is representing as number? I can't find the legends in this figure.

As for the machine learning issue, I have the same concern as other reviewers about overfitting. When a machine learning method is introduced for scientific research, there is a tradeoff between prediction and interpretation. If the author insists on a simple model, they should provide more interpretation about this model such as SHAP values. Fig.5 show ANOVA for those five amino acid levels, which is commonly used to explain linear model instead of KNN.

Version 2:

Reviewer comments:

Reviewer #2

(Remarks to the Author)

The authors have adequately addressed remaining inquiries from this Reviewer.

Reviewer #4

(Remarks to the Author)

All my comments have been addressed.

POINT BY POINT RESPONSE TO THE REVIEWER COMMENTS

Reviewer #1 (Bioinformatics in Cancer diagnosis):

General Comments:

In this paper, the authors have developed a novel immunodiagnostic platform to detect and identify a panel of amino acid residue biomarkers that provide a signature of cancer-specific immune activation associated with tumor development and distinct from autoimmune and infectious diseases, while being conveniently measured optically in neat patient blood plasma.

By simply measuring the total concentrations of cysteine, free cysteine, lysine, tryptophan, and tyrosine protein-incorporated biomarkers and analyzing the results with supervised machine learning, they can identify 86% of cancers with almost 0% false positive rate (N=92).

They also demonstrated that the cancer, healthy, and autoimmune/infectious biomarker pattern changes are all statistically significant with p value smaller than 0.0001. In their study the smaller scale changes in biomarker concentrations reveal inter-patient differences in immune activation which predict response to treatment. Specific concentration ranges of cysteine, free cysteine, lysine, tryptophan, and tyrosine biomarkers are shown to be predictive of response to Cyclin- dependent kinase inhibitors (CDKi's) in advanced breast cancer patients (p-value smaller than 0.05) and allowed identification of 100% of responding patients (N = 30). Their findings highlight the potential of the systemic immune response in cancer diagnostics. Furthermore, differences in amino acid biomarker space provide a cross sectional view of the biochemical phenotype differences between disease states providing new insights into central questions in cancer biology.

Specific comments (Statistical):

1. The use of Machine Learning techniques to identify the biomarkers is becoming common these days but identification of immunodiagnostic biomarkers indeed is very novel in this paper.

RESPONSE: We thank the reviewer for highlighting the novelty of our immunodiagnostic biomarkers and our method of identifying them.

2. Since this is a high dimensional problem, the designing the experiment is very tricky. But the authors of successfully designed the experiment by using a novel biological embedding technique.

RESPONSE: We thank the reviewer – and we agree that the use of an embedding technique to identify biomarkers is novel and holds great potential in the field.

3. The authors have performed a very sophisticated data analysis to achieve their goals.

RESPONSE: We thank the reviewer, and indeed we endeavoured to support our machine learning calculations also with classical statistics.

4. MANOVA to detect multi cancer early detection is very effectively done done for clustering the types.

RESPONSE: We thank the reviewer for MANOVA has been effectively applied to our data analysis.

5. The classification using cross-validation sample is also done vey effectively.

RESPONSE: We thank the reviewer also for considering the cross-validation well executed.

6. Overall, the results are also very significant and the study is very informative, novel and the state-of-the art.

ACTION: We thank the reviewer for the excellent comments and suggestions about our work, and in particular highlighting the innovative and robust elements in our biomarker design and statistical and machine learning validation.

Reviewer #2 (cancer FA metabolism, biomarker):

Title: Immunodiagnostic plasma amino acid residue biomarkers detect cancer early and predict treatment response

In the current study, Tang and colleagues established a panel of amino acid residue biomarkers for cancer-specific immune activation that is associated with tumor development and that is distinct from autoimmune and infection disease. By measuring total concentrations of cysteine, free cysteine, lysine, 38 tryptophan, and tyrosine protein-incorporated biomarkers, coupled with supervised machine learning, authors were able to identify 86% of cancers (N=92) at 100% specificity. Smaller scale changes in biomarker concentrations showed inter-patient differences in immune activation that predicted response to CDKis in the context of breast cancer.

Overall, the study is of considerable interest.

RESPONSE: We thank the reviewer for recognising the broad interest of the study.

However, there are several concerns.

RESPONSE: We thank the reviewer for highlighting these points. We believe we addressed all concerns below, which in our opinion, helped improving significantly our work.

1- Subject characteristics. A table describing subject characteristics among the comparative groups would be more easily interpretable compared to the excel file that has several tabs. There are also issues with reported DOB (for instance, healthy control #17 has a DOB of 31/01/1957 with a collection date of 20/06/2017. Assuming these were typos, age distributions among cases and controls appear quite different. Is there an association between reports biomarker values and patient characteristics, such as age and sex? Additionally, were patients fasted when blood was collected? Such information is important to assess for potential confounders.

RESPONSE:

We reported the DOB with the format DD/MM/YY, hence the 31/01/1957 DOB and 20/06/2017 collection date is not a typo. However, we've updated the DOB with clearer written and also included the "Age at collection" for easier visualization. (Highlighted in Supplementary Data_Patient clinical and demographic characteristics information_revised)

We provide below the age distribution of cases and controls, showing that the cancer and healthy patients were approximately age matched.

We analysed our measured Amino Acid Concentration Signature (AACS) for any dependence on patient age, and confirm that this is not a confounding factor. Patients < 50 and > 50 years of age without cancer had comparable AACS, as shown below.

Patients < 50 and > 50 years of age with cancer had comparable AACS, as shown below. (Note: CR represents Cys_R, C stands for Cys_T, Y denotes Lys, W signifies Trp, and Y also refers to Tyr. The manuscript uses the latter notation, while the single-letter version is shown here for simplicity.)

Furthermore, we have analysed our measured AACS for any dependence on patient biological sex and confirm that this is not a confounding factor. As shown below, male and female healthy patients had comparable AACS.

Additionally, male and female cancer patients had comparable AACS.

Regarding fasting, for the cancer screening indication, we did not require fasting before blood was collected. This is because the intended use environment of our primary test is a cancer screen, and we have been advised by our clinical partners to not require fasting because this would limit screening appointments to the morning, causing a bottleneck. For the CDKi treatment response prediction, patients were fasted before blood was collected, because they were receiving other blood work on the same day which did require fasting. They overlapped with the distribution of the non-fasting breast cancer patients.

2- Cancer staging- based on the supplemental table, it appears that most cancers are advanced stage (for instance, for PDAC all but 1 are stage III-IV). Thus, utility of the test multi-cancer ‘early’ detection is unclear and not supported by provided data.

RESPONSE: We thank the reviewer for highlighting this aspect. We were limited in accessing early-stage samples by typical late cancer presentations, a particular issue in PDAC as the reviewer highlights. However, we have performed additional experiments and following recruiting more early-stage PDAC patients. These experiments confirmed the trend discussed throughout the manuscript, of detectable, strong signals for early-stage cancer stemming from the mechanism of action of detecting the host immune response to tumour development. The figure below now replaces Figure 4g in the text.

We would also like to highlight that, following the present proof of concept study, we have progressed to a publicly funded clinical validation study (MODERNISED) towards regulatory approval(1). In this study, we will prospectively recruit N=1000 cancer patients and N=350 controls, aiming for an equal stage distribution among 10 cancer types. This study, which will complete in 2 years until 2026, will follow this current manuscript and will be submitted to a medical journal.

3- While this reviewer appreciates the innovative approach, the cohort size is too small to make any definitive conclusions regarding utility of the test for MCED or for predicting response to treatment.

RESPONSE: We appreciate the reviewer's recognition of the innovative nature of our approach. As this is a proof-of-concept study, our primary objective was to demonstrate the feasibility of the PCS technology in distinguishing between cancer and healthy samples, and our secondary objective was to evaluate feasibility of predicting response to treatment.

We would like to highlight that in response to reviewer feedback, we have now increased our total samples measured for this proof-of-concept study to N=170. This has now been clarified in the main text.

We are not seeking to demonstrate clinical utility at this stage, and doing so would be outside of scope of the present work. We would like to highlight that diagnostics development follows the timeline of proof of concept, clinical validation, and clinical utility. Demonstration of utility, which requires large screening trials as well as health economic analysis and collection of real-world data on clinical outcomes, usually occurs 5 years after completion/publication of the initial proof of concept work (the present study). And we're on track of doing this.

4- Although authors include patients with autoimmune and infection, other benign conditions pertinent to the field of cancer diagnosis were not included (e.g. benign polyps or low-grade dysplasia cysts, chronic pancreatitis, etc), precluding evaluation of specificity towards cancer.

However, this is not a major demerit given challenges associated with acquisition of such specimens.

ACTION: We thank the reviewer for these excellent suggestions and appreciate their recognition of the challenges involved in obtaining specimens from various benign conditions relevant to cancer diagnosis. We are currently conducting an additional study that includes a broader spectrum of samples, such as benign nodules and other conditions pertinent to differentiating benign and malignant disease.

This follow-up study aims to further evaluate the specificity of our AACCS technology toward cancer, providing more comprehensive insights into its diagnostic utility. We look forward to sharing these results in future publications.

5- Classifier performance. Although authors used a validation set, these were based on the same cohort. As such, any potential confounder would be present in both the training and validation set. Additionally, see comment 2 above.

RESPONSE: We thank the reviewer for highlighting our use of a validation set. Given our small sample size for cancer detection (N=92, N=97 with additional samples added at review), we used the industry-standard method of 5-fold cross validation to ensure that the classifier is validated on data on which it is not trained, as shown in Figure 4c. 5-fold cross validation partitioned the data set into 5 folds, and within each fold 80% of the data was used for training whilst 20% was unseen and held-back for validation. The classifier performance was assessed on the unseen 20% validation set within each fold. The held back 20% validation set was sliding, such that averaging the validation performance across the folds reveals the accuracy with N=92/97 datapoints ultimately used for validation as an unseen set.

We felt that given the small sample size (as the reviewer has highlighted), industry standard 5-fold cross validation was more appropriate than partitioning the data (e.g., 75-25) for hold-out validation, and acknowledge that therefore the training and validation sets use the same cohort. The reviewer highlights the resulting impact of possible confounding factors. However, point out that we performed additional analysis on subject characteristics in our response to the reviewer's point 1 and confirmed that potential confounders the reviewer highlighted of biological sex and age did not affect this cohort.

In our ongoing clinical validation study (MODERNISED), patients will be recruited across 5 geographic centres, minimising the risk of any potential confounding factor impacting a single recruitment site.

6- Figure 5- the number of patients that did not respond is very low (n= 5). Of note, the supplemental table indicates 4 non-responders rather than the 5 that are depicted in Figure 5C. Moreover, it does not appear that there is a great degree of separation, except for one individual. As mentioned above, the sample size is too small and subject to overfitting from models. Additionally, these analyses do not factor in patient and tumor characteristics (e.g. stage) that may be associated response rates.

RESPONSE: We thank the reviewer. We accepted the reviewer's comments that the number of non-responding patients was low, and thus we prospectively recruited N=3 additional non-responding patients and performed additional experiments to address this. The updated Figure 5c is below and included in the revised manuscript. We agree that this makes the separate

distribution (line) of non-responding patients, in the lower left-hand portion of the graph, more clear.

This increases our predicting treatment response cohort size to N=33, and gives us more confidence in the conclusions of the machine learning models, again trained and validated using industry standard and sample-size-appropriate approaches such as 5-fold cross validation.

We thank the reviewer for highlighting potential confounders in cancer stage or tumour characteristics. The class of drugs we are predicting response to, cyclin dependent kinase 4/6 inhibitors, are currently approved for the treatment of HR+, HER2- Stage 4 cancer patients. Therefore, all patients in this cohort were Stage 4 and had HR+, HER2- tumour characteristics, so variation in tumour stage or molecular characteristics would not have impacted our response rates.

7- Minor: Pancreatic cancer is not a prevalent cancer.

RESPONSE: While we acknowledge that pancreatic cancer is less prevalent compared to other cancers, its clinical importance is substantial due to its typically late diagnosis and poor prognosis. Early detection of pancreatic cancer remains a critical unmet need in oncology, as early-stage diagnosis significantly improves treatment options and survival outcomes.

By focusing on technologies that have the potential to detect pancreatic cancer at an early stage, our study addresses a pressing need in cancer diagnostics and aims to contribute valuable insights toward improving patient outcomes in this challenging area.

Reviewer #3 (metabolism in immune cells):

In this study, the authors demonstrate a method to detect cancer at an early stage by measuring amino acid residues in plasma as biomarkers. They have developed a simple, useable workflow, leading to accurate predictions, which is striking. The use of the method to stratify patient groups is a particular highlight.

RESPONSE: We thank the reviewer for highlighting our approach as striking, and pointing out our simple, useable workflow. We have developed the approach with accessibility and global scalability in mind.

However, I have a few questions about the method, the strength of some of the data, and its utility in particular situations.

RESPONSE: We thank the reviewer for highlighting these points and respond below.

1. It is not clear exactly how the authors settled on the final five amino acid residues to be tested. In Figure 2, they describe the composition of plasma, but in 2c, they do not consider the contribution of “c. 3000 other proteins”. Were these proteins excluded from the search for target residues, in order to focus more on immunoglobulins as a readout of immune changes?

- How the numbers of amino acid residues change in these other proteins should be shown.
- How did the authors decide what constituted a substantial change in the number of amino acid residues? For example, reduced cysteine varies in number from 0 to 1, and disulfide cysteine is generally in the range 20 – 35 (other than 80 in IgM). What is the normal variability for the number of each amino acid residue in this set of proteins?

RESPONSE: We thank the reviewer for the helpful clarifying questions. We believe this would be generally helpful and have therefore mirrored the discussion below and figures in the Revised Extended Data Figures.

Because our approach captures all targeted residues across all proteins present in blood plasma, our signal for each targeted amino acid type comprises a weighted average of the number of amino acids of this amino acid type in the protein sequence of each of the $i=1:3000$ proteins within blood plasma, weighted by the relative concentration of the i^{th} protein within the total protein content of plasma.

We considered all proteins within the search for target residues, as explained below. First, we analysed the protein sequences of all 3000 proteins within the human plasma proteome, and determined the impact of the number of targeted amino acids on the uniqueness of protein sequences if the order of the amino acids was, or was not, included in the identification. We determined that if the order was included – such as in the protein fluorosequencing work of Swathimani et al(2) – then only 2 amino acid types were required for uniqueness. If order was not included, as in our method, then 5 amino acid types were included for uniqueness (**Extended Data Fig.1**).

Extended Data Fig.1

Given that there are 20 canonical amino acid residues types and 5 distinct amino acid residue types are required for uniqueness, there are 15,504 possible biomarker combinations ($20 \text{ choose } 5$).

To determine the optimal combination among synthetically accessible residues, we considered possible combinations of amino acids and how these combinations contributed to uniqueness of protein sequences, given that we were aiming to capture maximum information from the combined signal across all amino acid sequences. The results are shown below as **Extended Data Fig.2**, for the case “without concentration bounds”, where simply the protein sequences rather than the expected healthy bounds of their concentration within plasma is considered.

Extended Data Fig.2

We then considered for the high uniqueness couples – including and to the right of “C, K, W” – whether we could devise a strategy to uniquely target these amino acid types chemically. For example, the highest differentiation combination “Q, L, K, V” would require chemically and quantitatively labelling Leucine and Valine which is not possible via current chemistry. We determined that “C, K, W, and Y” would be possible for us to quantitatively chemically label due to the specific reactive chemical nature of those R-groups.

The contributions of proteins to our composite biomarkers are weighted based on their relative concentration within plasma, as explained in the **Extended Data Fig.2**, Equations 1-2. Therefore, in Figure 2 we highlighted key proteins that comprise 96% of the signal (in healthy patients), hence the focus on immunoglobulins and albumin.

Information on the quantitative variability of cysteine, lysine, free cysteine, tryptophan, and tyrosine residues within different antibodies within the IgA, IgD, IgE, and IgG subclasses is limited due to the vast diversity of the variable region of heavy and light chains.

2. While the authors discuss that they want to examine amino acid residues in proteins, rather than free amino acid metabolites, it is not clear that they have taken any steps to address this in their biological processing. The plasma proteins are denatured during the labelling step (Fig. 3a), so how are free amino acids in plasma excluded from labelling?

RESEPNSE: Thank you for the clarifying question. We are not seeking to exclude free amino acid metabolites from our composite signature, as doing so would require an isolation step which is likely to disturb the proteins present and compromise quantification. However, free amino acid metabolites generally comprise less than 1% of the proportional signal once the proteins are denatured.

3. It is not clear what point the authors are trying to make in Extended Data 3 – the amino

acids shown change similarly in the different conditions, while the authors seem to suggest that the signals are different.

RESPONSE: We thank the Reviewer. We agree that this presentation is not particularly informative and have replaced this in Revised **Extended Data Fig. 5** with a statistical analysis of the importance of each amino acid type to the signals. The contribution of each included Amino Acid type to the signal is indicated by the F-Statistic in the figure below.

ANOVA Results: Log-Transformed Importance of Amino Acid Types to Cancer Status

Extended Data Fig.5

Interestingly, in differentiating between cancer and non-cancer immune activating conditions, the Trp (W) and Cys_T (C) amino acid types also contribute meaningfully to the signal, which are now shown as **Extended Data Fig. 10**.

ANOVA Results: Log-Transformed Importance of Amino Acid Types to Cancer Status vs Healthy vs Non-Cancer Immune

Extended Data Fig.10

4. Fig. 4g needs more data points to be convincing.

RESPONSE: We appreciate the reviewer’s suggestion, and in response, have recruited further pancreatic early-stage cancer patients allowing us to add more data points to **Figure 4g** in the manuscript and below. The results now show convincingly that we detect strong signals for early-stage cancer, we note in particular that all Stage I and II cancer patients are well separated from the healthy distribution.

5. It would be great if the authors had a particular piece of biological evidence pointing to the immune system being the main mediator of these effects. For example, does this method fail in patients that are already immunocompromised due to disease or drug treatment, and produce less immunoglobulin?

RESPONSE: We thank the reviewer for this excellent suggestion. To address this, we have now performed additional experiments and added another panel as new **Figure3e** to evidence the immune system being the main mediator of these effects. If our PCS signals are driven primarily by changes to the host immune response, then we would expect to be able to detect the difference between immunologically hot and immunologically cold tumours. To test this at the proof-of-concept level, we constructed a mouse model of hot and cold tumours and showed that these produce distinct PCS signals.

Reviewer #4 (Bioinformatics in Cancer diagnosis):

Manuscript by Tang et. al is about a diagnosis method for cancer and prediction of treatment response based on plasma amino acid residue in terms of immunosurveillance. Authors set up an optical detection with fluorogenic labels for immunoglobulin components considering 5 amino acid residues. They applied this method to detect multi Cancer and introduced a discriminant classifier for prediction, as well as an KNN classifier for cancer location prediction. Additionally, they applied this method to evaluate the CDKi cancer treatment considering the difference of sensitivity of patients. The major advantage of this method is the better sensitivity for early diagnosis compared with ctDNA methods.

RESPONSE: We thank the reviewer for highlighting the importance of our improved early-diagnosis sensitivity compared to ctDNA based methods.

Author should address the following concerns for publication purposes.

The first issue is about the selection of embedded amino acids. Out of 20 major amino acids, authors selected 5 types based on biological embedding. However, the rationale to select those amino acids is not clear in figure 2c and the authors claimed "We identified N=5 amino acid types whose numbers changed substantially across the proportion-weighted fractions, such that they would be detectable in an average biological embedding signature." Author should also provide the information of other amino acids to show a comprehensive selection. In the method part, authors introduced an embedding formula as amino acid number weighted protein concentration. How many proteins are considered for this calculation? Which plasma proteome database is used?

RESPONSE: We thank the reviewer for this helpful question. We have already provided information on the other amino acids, including the number of required amino acids (5) and choice of possible combinations of 5 biomarkers in response to Reviewer 3, Question 1. Regarding the number of proteins we considered in the proportion-weighted fraction calculation – we analysed 453 proteins predicted to be actively secreted to peripheral blood by the Human Protein Atlas and which had been detected via immunoassay, as these had a provided protein concentration which allowed us to calculate the proportion-weighted fractions. We then accessed the mapped protein sequences via UniProt and counted the number of amino acids of each amino acid type using Python ProtParam count_amino_acids.

As for the fluorogenic labels, authors considered confounding issues. It seems authors performed a positive/negative control in figure 3b and 3c to show the amino acid type is only related to the autofluorescence. Since the dye is only reactive to certain amino acids, I wonder if the authors also checked the calibration curve of amino acid with fluorescence and added this curve to figure 3c. Their slope should be similar and if not, the authors should consider matrix effects for this method.

RESPONSE: We thank the reviewer and wanted to clarify that in Figure 3b and 3c we showed that the fluorescence from the labelled amino acid type is only related to the concentration of the amino acid type, and not to any autofluorescence from the blood plasma itself. The provided slopes/functions thus provide the relationship between the fluorescence of the labelled amino acid type and its concentration in the solution.

After the samples containing the labelled amino acid type are background corrected to remove any minimal fluorescence from the label without the sample (the labels only become fluorescent after reaction), and further background corrected to remove any autofluorescence from the sample itself without the label, then the slopes/functions are used to calculate the concentration of the amino acid type in the solution from its labelled fluorescence.

We have clarified **Extended Data Fig. 3** to highlight the distinct fluorescent species which we measure for each labelled amino acid type.

Extended Data Fig.3

Authors selected MANOVA to show the distribution between case and control and then employed machine learning algorithms for prediction. A regular machine learning or statistical learning process should use dimension reduction methods such as PCA and UMAP to show the separation of samples. However, this work always uses a 3D visualization with amino acids to show the separation, which is not clear in figure 4f and 4d. If the amino acids are selected appropriately, I would expect a clear separation on PCA or UMAP.

RESPONSE: We thank the reviewer, and have now calculated the UMAP representations of both the (1) cancer versus healthy classification as well as (2) cancer versus healthy versus infection versus autoimmune. The figures are below and have been added as **Extended Data Fig. 8 and 9**.

Extended Data Fig.8

Extended Data Fig.9

As the reviewer anticipates, we do observe clear separation of the cancer versus healthy samples on the UMAP, selecting that we have chosen our biomarkers appropriately as the reviewer points out.

Furthermore, the UMAP visualisation of cancer versus healthy versus autoimmune versus infection shows that although distinct, cancer is more similar to an autoimmune condition than an infectious disease, which lends further support to our conclusion that the immune system regards cancer as more similar to “self” than “other”.

Though a 5 fold cross validation is performed for training, the machine learning algorithms are not evaluated for real test data, which should not be used for training process and I will concern the overfitting issues. The machine learning algorithms for prediction of cancer location and type are different and I can't find the model selection process. I would suggest the authors use automatic machine learning tools such as autogluon for this dataset and the embedding of different models might further improve the results. The interpretable machine

learning module of such tools will also tell the importance of different amino acids for different applications. Those results might give hints on the mechanism of immune surveillance for cancer.

RESPONSE: We thank the reviewer. **At the reviewer’s suggestion, we have performed an ANOVA analysis to analyse the contribution of each amino acid type to differentiation between cancer, healthy, and non-cancer immune controls and have added this as Extended Data Fig. 5 and 10.**

5-fold cross validation is an industry-standard method for preventing overfitting by ensuring that the model is validated on data it is not trained on, as described in Figure 4c. 5-fold cross validation partitioned the data set into 5 folds, and within each fold 80% of the data was used for training whilst 20% was unseen and held-back for validation. The classifier performance was assessed on the unseen 20% validation set within each fold. The held back 20% validation set was sliding, such that averaging the validation performance across the folds reveals the accuracy with N=97 datapoints ultimately used for validation as an unseen set.

However, we have now adopted an additional approach to address the reviewer’s suggestion to provide “real” (meaning held out) test data on which the model was not trained, and evaluated performance on a held-out 25% test set. As shown below and in **Extended Data Figure 6**, we divided the data set into an N=73 training/validation set and evaluated performance of the classifier on the validation set using 5X cross validation, achieving an AUROC of 0.95 as shown in **Extended Data Fig. 6a**.

Extended Data Fig. 6

We then evaluated the performance of the trained and validated classifier on a N=24 held-out independent test set, and observed an excellent **AUROC of 0.99** as shown in **Extended Data Fig. 6b**.

We have further considered Leave-Out-One Cross-Validation, in which a single datapoint is removed from the training and validation set, and used for testing. This is repeated N=97 times so that each point has been used for testing once, and not for training and validation. The resulting performance metrics for our linear Support Vector Machine (SVM) model are below and have been added as **Extended Data Fig. 7**.

Extended Data Fig.7

Overall, we observed strong performance across validation approaches, so we are confident that our machine learning classification is revealing real differences between the datasets.

Regarding the model selection process for cancer location/cancer type (this is a single outcome), we evaluated a random forest, support vector machine, discriminant, Bayesian, neural network, and weighted K-nearest neighbour classifiers on the N=77 cancer positive datapoints using 5X cross validation as above, and chose a weighted K-nearest neighbour because it achieved the highest accuracy in the validation set. 10 neighbors were considered with weights provided by the squared inverse of the Euclidean distance.

We thank the reviewer for the suggestion to use AutoML tools such as AutoGluon. Although AutoML tools provide an automated machine learning process, they are more suitable for commercial and industrial applications, where automation and efficiency are more important. However, in scientific research, reproducibility is crucial, and the automated process of AutoML has more randomness, which affects reproducibility. In addition, the AutoML framework is larger and more complex, while for our research, the standard machine learning process is sufficient and manually constructing a standard ML process is more controllable and reproducible while avoiding unnecessary complexity. Given that our dataset is relatively small, basic ML models such as SVMs is already proven to be widely used in health fields and sufficient(3), while automatic model selection such as deep learning or ensemble methods not only increases computational complexity, but may also lead to overfitting rather than actual performance improvement.

Although selecting models may enhance the results, ML process optimization is another research question that is beyond the scope of our current research. Our research focuses more on the possibility of disease screening by amino acid concentration detection and analysis, while ML algorithm development is not the priority. Also, given the large number of available ML pipeline tools and models, testing every tool in this research is neither practical nor meaningful. However, ML process optimization is still our next research direction. Future studies may explore more advanced strategies, especially algorithms to enhance performance and model interpretability, which can provide a deeper understanding of the role of amino acids in cancer immune surveillance.

Another important issue is about the discussion of biomarkers. It seems the core of this method is the selection of the 5 amino acids types and the authors didn't discuss why they can work. The overall hypothesis is about the immunology response of cancer and the reader might still doubt why 5 amino acid types are good and enough to stand for immunosurveillance. Meanwhile, those amino acids should also be covered by proteomics and metabolomics analysis and authors should cross check their results for support or disparity in their results. I will also suggest the authors try all of the 20 amino acid labeling to build a comprehensive residue model for cancer markers and the residual labeling technique might also be used for proteomics studies.

RESPONSE: We thank the reviewer for these thoughtful comments and for highlighting the importance of biomarker selection. We have explained the bioinformatic approach which guided our biomarker selection in response to Reviewer 3. Briefly, this involved calculating the number of amino acid biomarker types considered to provide a differentiable signal - which we found to be five amino acid types - then examining the information content provided by possible groups of amino acid types. This allowed us to conclude that the Cysteine, Free Cysteine, Lysine, Tryptophan, and Tyrosine Amino acid types would be sufficiently informative.

Our experimental approach relies on biorthogonal, fluorogenic, quantitative labelling - this means that to be suitable for our assay, an amino acid type R-group must be able to be site-specifically labelled with a dye which generates fluorescence only upon reaction with that amino acid type, which reacts exclusively with that amino acid type and not other amino acid types, and which generates fluorescence which is quantitatively related to the concentration of that amino acid type in the solution.

At this time, it is not feasible to label all 20 amino acids simultaneously, as existing techniques do not support precise and stable labelling across such a broad range of residues. For example, site-specific labelling tools for hydrophobic residues such as alanine, glycine, leucine, and isoleucine are not currently available within organic/synthetic chemistry.

We thank the reviewer for the suggestion to carry out a proteomic analysis of the indicated amino acids and clarify that indeed we have already carried this out. In existing Figure 3d, we used mass spectrometry data to predict the concentrations of each amino acid residue type within patient plasma samples. We then measured N=20 additional patient plasma samples and compared the results. As shown in Figure 3d, the experimental and theoretical results aligned, further supporting our selection of biomarkers and core hypothesis that these biomarkers are sufficient to simultaneously provide a signal from all proteins within patient plasma. We have clarified the main text to explain that this analysis uses existing proteomic data from the Human Protein Atlas.

The last concern is about the figures. Some figures are not necessary to provide information such as figure 1d, 2a, and 4c. Those information are either hard to read or background knowledge from the textbook. Meanwhile, the 3D scatter plot is hard to read for cluster separation.

RESPONSE: We thank the reviewer. Whilst we acknowledge the reviewer's point, we note that the readership of Nature Communications is diverse, and each queried figure addresses a point raised in the review process. For example, Figure 1d explains the proteins which

contribute substantially to our Protein Cross Section signals, beyond simply the immunoglobulin components highlighted in Figure 2c. Figure 2a explains how a machine learning embedding works, and whilst textbook knowledge for readers familiar with machine learning, we believe this would be important for other readers of a more experimental background. Similarly, Figure 4c explains the process of K-fold cross validation, which we believe would be important for readers not familiar with machine learning.

We have, however, added the UMAP requested by the reviewer as **Extended Data Fig. 8 and 9** to assist with cluster analysis.

REFERENCES

1. Southampton Clinical Trials U. MODERNISED Trial Overview: University of Southampton; 2025 [updated 2025/03/07. Available from: https://www.southampton.ac.uk/ctu/trialportfolio/listoftrials/modernised.page#trial_overview
2. Swaminathan J, Boulgakov AA, Hernandez ET, Bardo AM, Bachman JL, Marotta J, et al. Highly parallel single-molecule identification of proteins in zeptomole-scale mixtures. *Nature Biotechnology*. 2018;36(11):1076-82.
3. Guido R, Ferrisi S, Lofaro D, Conforti D. An Overview on the Advancements of Support Vector Machine Models in Healthcare Applications: A Review. *Information [Internet]*. 2024; 15(4).

REVIEWER COMMENTS

Reviewer #1 (Bioinformatics in Cancer diagnosis):

Early detection of cancer biomarker to predict treatment response.
The revision is satisfactory in terms of statistical contents.

RESPONSE: We thank the reviewer for their helpful comments which aided our revision.

Reviewer #2 (cancer FA metabolism, biomarker):

The authors were largely responsive to prior comments.

RESPONSE: We thank the reviewer for their helpful prior comments.

Nevertheless, there are a few additional requests.

RESPONSE: We are happy to address these additional requests below.

1) Please include a table that summarized patient characteristics described in individual tabs of the supplemental materials.

RESPONSE: We have now provided a summarized patient characteristics tab in the revised demographic and clinical information table within the supplemental materials, “Supplementary Data_Patient clinical and demographic information and summarised.xlsx”

2) This Reviewer appreciates sub-analyses showing association between measured AAs and age (stratified by <50 and >= 50). However, these plots are based on a log10 scale. It would be of benefit to show correlation matrices between AA levels and age as continuous variables for cases, controls and the entire cohort. For classifier performance, it would be of benefit to include age as a co-variable.

RESPONSE: We thank the reviewer for noting the sub-analyses showing association between measured AAs and age.

We have now provided correlation matrices between AA levels and age as continuous variables for cases, controls, and the entire cohort. The analysis below, now added as **Extended Data Fig. 11**, reveals that no amino acid type is strongly correlated with Age in either the Cases, Controls, or All Subjects. The K and CF amino acid types are weakly/moderately negatively correlated with Age in the Cases, while the W and Y amino acid types are weakly positively correlated with Age in the Controls.

To further explore whether these weak correlations could impact cancer status prediction, we have now additionally considered age as a co-variable for classifier performance. As shown below, and added as **Extended Data Fig. 12**, when we train the same classifier (Ensemble Subspace Discriminant) for cancer

detection as presented in Figure 4b, the AUROC remains comparable (0.95). Sensitivity at 100% specificity reduces from 78% to 61% when age is included, supporting that the addition of age does not improve classifier performance and may introduce minor overfitting.

Collectively, the additional analyses support that age is not a confounding factor in cancer status prediction using our novel amino acid residue biomarker approach.

Reviewer #3 (metabolism in immune cells):

The authors thoughtfully addressed my comments and both clarified and strengthened key points of the paper. I recommend this work for publication.

RESPONSE: We thank the reviewer for their helpful comments which strengthened the paper.

Reviewer #4 (Bioinformatics in Cancer diagnosis):

The authors have addressed my concerns on selection of amino acid type.

RESPONSE: We thank the reviewer for their helpful comments and suggestions.

1) My second concern is not responded. I would like to know if pure amino acids instead of residual of protein can be labeled and show the same calibration curve. This analysis will get rid of the specificity issue of this methods when the calibration curve of amino acid show the same slope as protein.

RESPONSE: We thank the reviewer for clarifying their concern, which originally, we had not correctly understood.

Yes, it is indeed possible to label pure amino acids instead of amino acid residues within proteins to determine if they have the same slope as proteins within the calibration curve. In fact, we have carried out this analysis when we were developing the underlying labelling chemistries^{1,2}. For example, Figure 2b of our 2015 Nature Chemistry paper reporting the lysine chemistry showed that the amino acid L-Lysine had the same calibration curve as a variety of protein standards, including those (BSA, β -Lactoglobulin, and Lysozyme) included in the present study.

Because the previously published work included lysine and cysteine, we have performed additional experiments exploring whether Tryptophan and Tyrosine free amino acids had the same slope as proteins containing these residues, and we confirm this below. The “BSA_p”, “Lys_p” and “Beta-Lac_p” series are free Tryptophan and free Tyrosine amino acids at a series of equivalent concentrations to the protein standards BSA, Lysine, and Beta-Lactoglobulin. The protein-incorporated amino acid residue curves have the same slope as the free amino acids. The linear shift upwards is expected because the fluorescence intensity of tryptophan residues is known to be increased in proteins relative to the free amino acid due to hydrophobic environment effects³.

2) The third concern has been addressed by UMAP.

RESPONSE: We thank the reviewer for noting that we have now provided UMAP representations of our data.

However, I found 3D figures in this paper showed different x, y, z axis. There are 10 combinations to select 3 axis from 5 amino acids types and 3D figure could only show 3 amino acids. If the purpose for this visualization is show the separation of the amino acid selection, 3D figures will not get rid of the concern of cherry-picking.

RESPONSE: We thank the reviewer. Because representing high dimensional data using UMAP or a subset of features each has benefits and limitations, the paper includes both representations for a holistic view. While UMAP presents information from all (5/5) dimensions, avoiding the concern of cherry-picking, it suffers from a lack of interpretability and variable reproducibility.

In our case, interpretability is important, and we only have 5 variables, so we feel that visualising real concentrations is meaningful. For 3D visualisation of cancer status prediction (**Fig. 4a** and **Fig. 4c**), we have chosen the 3/5 features (Cys_R, Lys, Tyr) which contributed most to healthy versus cancer class separation in the ANOVA analysis in **Extended Data Fig. 5**.

For the 2D visualisation of cancer versus healthy versus autoimmune versus infection controls (**Fig. 4g**), we have chosen the 2 dimensions (Cys_R, Cys_T) which contributed most to this class separation in the ANOVA analysis in **Extended Data Fig 10**. We have clarified these points in the text.

To ensure consistency in feature selection approach throughout the manuscript, we have now further adopted the same approach in **Fig. 5**, replacing old **Fig. 5c** with a new panel showing the 3 dimensions of information with highest Feature Ranking importance using ANOVA (revised plot shown below).

We have added the Feature ranking plot shown below as **Extended Data Fig. 13**.

3) Meanwhile, the new figure 4g looks confusing. What's the meaning of the shadow? Why some dot is representing as number? I can't find the legends in this figure.

RESPONSE: We thank the reviewer. All PDAC data points in **Fig. 4g**, shown as dots, are represented also with a number which is the clinical stage of that patient's cancer. The shadow is simply a schematic showing a clustering of metastatic cancer patients near the healthy distribution. We have clarified both points in the figure legend.

* We note that due to the addition of a new panel (**Fig. 4d**) to address the Reviewer's machine learning point 4 below, old "**Fig. 4g**" referenced by the Reviewer is now **Fig. 4h** in the main text.

4) As for the machine learning issue, I have the same concern as other reviewers about overfitting. When a machine learning method is introduced for scientific research, there is a tradeoff between prediction and interpretation. If the author insists on a simple model, they should provide more interpretation about this model such as SHAP values. Fig. 5 show ANOVA for those five amino acid levels, which is commonly used to explain linear model instead of KNN.

RESPONSE: We sincerely appreciate the reviewer’s insightful comment regarding the potential risk of overfitting and the importance of interpretability when using machine learning in biomedical research.

As the reviewer rightly points out, model interpretability is critical, particularly when applying machine learning in clinical or biomarker discovery contexts. In our study, we carefully selected linear models to alleviate overfitting concerns, given the limited sample size. However, we acknowledge the potential overfitting possibility and address it with more analyses as below. While ANOVA is conventionally used with linear models, in our analysis ANOVA was applied not as a model inference tool, but to statistically assess the differentiation of individual amino acid levels between cancer and healthy groups. We now clarify this distinction in the text to prevent misinterpretation (see updated introduction of ANOVA in the main text (pg. 10), **Extended Data Fig. 5 and 10**).

Furthermore, following the reviewer’s recommendation, we incorporated **SHAP (SHapley Additive exPlanations)** to enhance model interpretability and verify the robustness of our SVM predictions. SHAP enables feature-level attribution of model predictions, offering an interpretable view into how individual amino acid concentrations contribute to classification outcomes and has been widely used in biomedical research ⁴⁻⁶.

To guard against overfitting and assess the **stability of feature importance**, we performed **5-fold cross-validation** with stratified sampling and computed **normalized SHAP values** across all folds. We then examined whether the relative contributions of features remained consistent across splits. The **Friedman test** ($\chi^2 = 5.96$, $p = 0.201$) revealed no statistically significant differences in SHAP-based feature rankings across folds, suggesting that the model’s interpretation is not dominated by fold-specific artifacts.

We now include a new figure (**Fig. 4d**, shown below) visualizing the **mean normalized SHAP values \pm SD** for each amino acid feature. Notably, features identified by SHAP as having the highest model contributions (e.g., **Cys_R, Tyr, Lys**) closely mirrored those flagged as statistically significant by the ANOVA analysis (**Extended Data Fig. 5**, reproduced below). This convergence of evidence from both model-based and statistical approaches reinforces the biological relevance of these amino acids in cancer classification and underscores the robustness of our findings.

Extended Data Fig. 5

Similarly, we have performed additional SHAP analysis on the linear SVM model used to classify patient response to CDKi treatment. To ensure that our interpretation is not biased by specific data splits, we conducted five-fold cross-validation, computed normalized SHAP values for each fold, and examined the stability of feature importance.

To statistically test whether the ranking of SHAP-based feature contributions remained consistent across folds, we applied the Friedman test. The result ($\chi^2 = 5.60$, $p = 0.231$) indicates that there is no significant difference in feature rankings across folds, suggesting that the model interpretation is robust and not driven by fold-specific variation.

We now include a new panel (Fig. 5f, shown below) in the revised manuscript that presents the mean normalized SHAP values \pm standard deviation for each amino acid feature involved in the CDKi response prediction. This figure offers a transparent and data-driven explanation of the model's predictions.

We note that due to the addition of a new panel to each of Fig. 4 and Fig. 5, the figure referencing in the main text has been updated.

References:

- 1 Yates, E. V. *et al.* Latent analysis of unmodified biomolecules and their complexes in solution with attomole detection sensitivity. *Nature Chemistry* **7**, 802-809 (2015). <https://doi.org/10.1038/nchem.2344>
- 2 Hakala, T. A. *et al.* Accelerating Reaction Rates of Biomolecules by Using Shear Stress in Artificial Capillary Systems. *Journal of the American Chemical Society* **143**, 16401-16410 (2021). <https://doi.org/10.1021/jacs.1c03681>
- 3 Vivian, J. T. & Callis, P. R. Mechanisms of tryptophan fluorescence shifts in proteins. *Biophys J* **80**, 2093-2109 (2001). [https://doi.org/10.1016/s0006-3495\(01\)76183-8](https://doi.org/10.1016/s0006-3495(01)76183-8)
- 4 Lundberg, S. & Lee, S.-I. in *31st Conference on Neural Information Processing Systems (NIPS 2017)* (Long beach, CA, USA, 2017).

- 5 Fan, Y.-W. *et al.* Exploring kinase family inhibitors and their moiety preferences using deep SHapley additive exPlanations. *BMC Bioinformatics* **23**, 242 (2022).
<https://doi.org/10.1186/s12859-022-04760-5>
- 6 Shimazaki, T. & Tachikawa, M. Collaborative Approach between Explainable Artificial Intelligence and Simplified Chemical Interactions to Explore Active Ligands for Cyclin-Dependent Kinase 2. *ACS Omega* **7**, 10372-10381 (2022).
<https://doi.org/10.1021/acsomega.1c06976>